# Ternary molecular switching in a single-crystal optical actuator with correlated crystal strain

Jacqueline M. Cole [1,2,3,4] ✉, David J. Gosztola [3],
Jose de J. Velazquez-Garcia [1] & Jeffrey R. Guest [3]

A growing portfolio of single-crystal optical actuators is forging a new class of photonic materials that hold prospects for quantum technologies. Ruthenium-based complexes that exhibit this phenomenon via $SO_2$-linkage photo-isomerisation are of particular interest since they display multiple metastable states, once induced by green light; yet, complete photoconversion into each $SO_2$-isomeric state is rarely achieved. We discover a new complex, trans-[Ru($SO_2$)($NH_3$)$_4$(4-bromopyridine)]tosylate$_2$, that produces 100% photo-converted $\eta^1$-OSO isomeric crystal structures at 90 K, which fully transition into $\eta^2$-(OS)O photoisomers upon warming to 100 K, while the dark-state $\eta^1$-$SO_2$ structure is wholly recovered by heating the crystal to room temperature. Crystal structures and optical-absorption profiles of each state are captured via in-situ light-induced single-crystal X-ray diffraction and optical-absorption spectroscopy. Results show that both photoisomeric species behave as optical switches, but with distinct optical properties. The photoisomerisation process causes thermally-reversible micro- and nanoscopic crystal strain, as characterised by optical microscopy and in-situ light-induced atomic-force microscopy.

Single-crystal optical actuators are stimulating a new field of chemistry, given their wide-ranging prospective applications that range from light-driven molecular machinery[1–3], to microrobotics[4–7], to optical data storage[8], to futuristic quantum-computing circuitry[9]. Single crystals are a particularly attractive form of an optically actuating material since they offer a highly pure solid-state medium for photonic control. Crystalline actuators whose functional origins lie in light-induced structural changes are especially alluring because such changes exhibit long-range effects that propagate throughout a periodic crystal-lattice environment; thereby affording an ordered array of photonic signatures. These photostructural effects are often correlated, and 'out-of-equilibrium' in nature, a characteristic that has been coined as 'the real terra nova of solid-state chemistry'[10].

Single-crystal optical actuation has been demonstrated in a diverse range of organic, inorganic and organometallic compounds[11–26]. Coordination complexes that contain transition metals, such as ruthenium, are of particular interest since they exhibit metal-to-ligand charge transfer (MLCT) that gives rise to broad-band optical absorption, while maintaining good thermal stability. Linkage photo-isomerisation is often the molecular-scale phenomenon that underpins the microscopic manifestation of single-crystal optical actuation in coordination complexes[24]. We therefore sought to discover a new single-crystal optical actuator by developing an extended series of ruthenium-tetraammine complexes that undergo $SO_2$-linkage photoisomerisation[27–45]. Their generic formula is trans-[Ru($SO_2$)($NH_3$)$_4$(X)]$^{m+}$Y$_n$, where X is a trans ligand relative to the photoactive $SO_2$

[1]Cavendish Laboratory, Department of Physics, University of Cambridge, Cambridge, UK. [2]ISIS Neutron and Muon Source, STFC Rutherford Appleton Laboratory, Harwell Science and Innovation Campus, Didcot, UK. [3]Center for Nanoscale Materials, Argonne National Laboratory, 9700 South Cass Avenue, 60439 Illinois Lemont, USA. [4]Materials Science Division, Argonne National Laboratory, 9700 South Cass Avenue, 60439 Illinois Lemont, USA. ✉e-mail: jmc61@cam.ac.uk

ligand, Y is a counter ion and m and n are integers whose values are determined by charge balancing (hereafter, these will be known as [RuSO$_2$] complexes). Many variations of X and Y can support one or both η$^1$-OSO or η$^2$-(OS)O photoisomeric configurations within the single-crystal form, whose light-induced structure is metastable when kept suitably cooled (typically to 100 K) and which thermally reverts to its η$^1$-SO$_2$ dark-state crystal structure. Single-crystal optical actuation has been reported in one [RuSO$_2$] complex, to date, and adopts the form of a thermally-reversible macroscopic crystal-peeling effect[38].

Peeling is one of the more novel types of macroscopic crystalline deformations that may occur in complexes, owing to the untenable crystal strain that molecular-scale photostructural effects can impart: crystals may peel, bend, stretch, crack, fracture or explode, when subjected to light[11].

This peeling effect was deemed to stabilise the substantial but incomplete (52%) photoconversion level of the η$^1$-OSO isomer in [Ru(SO$_2$)(NH$_3$)$_4$(3-phenylpyridine)]Cl$_2$.H$_2$O[38]. A [RuSO$_2$] complex would ideally support 100% SO$_2$-linkage photoisomerisation so that the dark- (0) and light-induced (1,2,...) states can be readily distinguished as

integer encodings for photonic applications. Yet, until now, complete η$^1$-OSO photoconversion had only been reported in single crystals of two [RuSO$_2$] complexes[42,44], neither of which display macroscopic optical actuation. Furthermore, the η$^2$-(OS)O photoisomer that is often found in [RuSO$_2$] complexes has never been structurally isolated with 100% photoconversion; rather, it has been found to co-exist with one or both of the η$^1$-OSO photoisomer and η$^1$-SO$_2$ dark-state configurations[27–38,43,45].

This paper reports the discovery of a new [RuSO$_2$] complex, *trans*-[Ru(SO$_2$)(NH$_3$)$_4$(4-bromopyridine)]tosylate$_2$] (**1**, Fig. 1a) that exhibits a thermally-reversible 'cracking' form of single-crystal optical actuation. Thereby, we show that its η$^1$-SO$_2$ dark-state crystal structure undergoes 100% SO$_2$-linkage photoisomerisation to create a pure η$^1$-OSO photoisomeric crystal structure at 90 K or a pure η$^2$-(OS)O photoisomeric species at 100 K. Each photoisomer is metastable at their stated temperature and is accompanied by a substantial level of micro- and nanoscopic crystal strain, which diminishes as **1** thermally recovers its η$^1$-SO$_2$ dark-state crystal structure.

This demonstration of three distinct 100%-converted SO$_2$-linkage isomeric species in a [RuSO$_2$] complex is important because it reveals

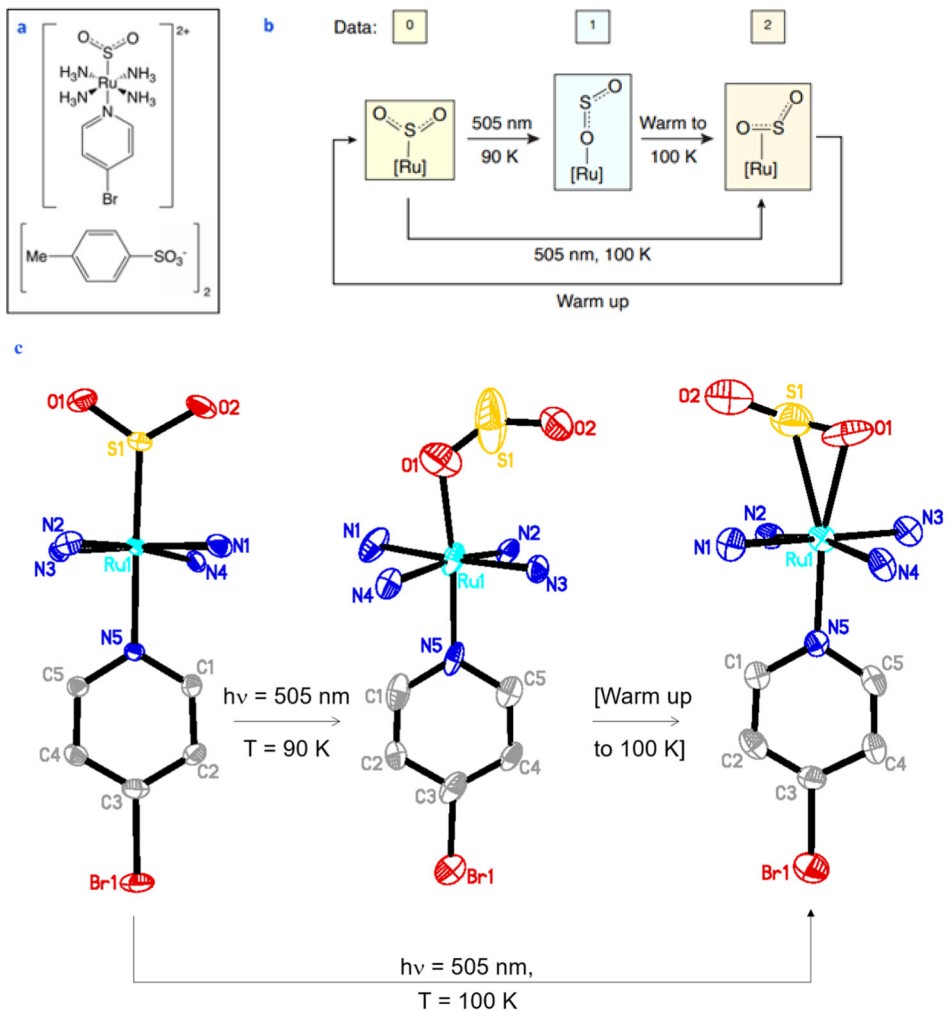

**Fig. 1 | Chemical Structure of 1 in its dark-state and photoisomeric forms. a** Chemical schematic diagram of **1**, whose (**b**) crystals can be light and temperature tuned to behave as a ternary molecular switch (Data: 0, 1, 2). The molecular origins of this ternary switching behaviour lie in photostructural changes within the crystal structure of **1**, caused by 100% SO$_2$ linkage photoisomerisation that occurs when the crystal is exposed to 505 nm light while maintained at low-temperature: the η$^1$-OSO or η$^2$-(OS)O photoisomer forms at 90 K or 100 K, respectively. The latter photoisomer can also be generated by warming up the former by 10 K. The η$^1$-SO$_2$ dark-state crystal structure is recovered upon warming up the crystal. **c** The asymmetric unit of the three crystal structures of **1** captured in each of its isomeric-state configurations. Anisotropic displacement parameters are shown at the 50% probability level. Hydrogen atoms and tosylate counterions are omitted for clarity.

the foundational possibility of creating ternary photonic signatures of high purity in single-crystal optical actuators (Fig. 1b). We now describe our findings through the structural and optical characteristics of **1** which we determined using in-situ light-induced single-crystal X-ray diffraction (also known as 'photocrystallography')[46–48], single-crystal optical-absorption spectroscopy and optical microscopy[41] and in-situ light-induced low-temperature atomic-force microscopy. These findings also show how its ternary molecular switching can be controlled by temperature and visible light (Fig. 1b).

## Results and discussion
### Dark- and light-induced crystal structures of 1
The dark and photoinduced crystal structures of **1** at 90 K and 100 K are displayed in Fig. 1c. Three fully-formed $SO_2$-isomeric crystal structures of **1** with long-range order were captured under the stated light and temperature conditions.

The $\eta^1$-$SO_2$ dark-state crystal structure is stable at 100 K, as judged by its modest anisotropic displacement parameters (ADPs) which are largely isotropic in form, save for the slight elongation of ADPs for terminal atoms that one would expect in crystallography. The octahedral coordination sphere of ruthenium supports a tetraammine-based equatorial plane with highly regular geometry, Ru-$N_{ammine}$ (2.126(3)-2.130(3) Å) and two perpendicular $SO_2$ and 4-bromopyridyl ligands, with Ru-$N_{pyridyl}$ (2.125(3) Å), Ru-S (2.1175(9) Å). The 4-bromopyridyl ligand will bear a *trans* influence on the strength of the Ru-S coordination, which in turn will impact the extent to which the $SO_2$ ligand can become labile for photoisomerisation. We would expect the *trans* ligand in **1** to exhibit a similar level of *trans* influence to that of its isostructural 4-chloropyridyl analogue[36] [whose Ru-$N_{pyridyl}$ is 2.122(2) Å; Ru-S is 2.113(1) Å], especially given that the pKa values of 4-bromopyridine (4.01) and 4-chloropyridine (3.98) are also similar[49]. The $\eta^1$-$SO_2$ isomer displays a regular geometry [S1-O1: 1.487(3) Å; S1-O2: 1.451(3) Å; O1-S1-O2: 115.2(2)°].

Having acquired crystallographic data for **1** under dark-state conditions at 100 K, the crystal was kept on the diffractometer at 100 K and exposed to 505 nm light for 2.75 h at 15 µW/mm², before collecting and analysing another dataset. This afforded a metastable light-induced crystal structure at 100 K which contains a 100% photoconverted, crystallographically ordered, $\eta^2$-(OS)O isomer. Such an $SO_2$-ligand configuration has never been captured in the crystal structure of any salt. Even in non-ionic complexes, few reported crystal structures display fully formed and ordered $\eta^2$-(OS)O isomers and these are coordinated to Rh or Mo, where the metal mostly resides in the zero oxidation state[50,51], except for a few instances[52,53].

An unusual physical observation was made during this experiment: the crystal of **1** displayed a flickering yellow-to-blue photochromic change during the first few minutes of its light exposure at 100 K, after which it essentially remained yellow in colour. The internal crystal temperature was still settling during these few minutes, as the crystal had only just reached 100 K. Given that metastable photochromism tends to be observed when metastable $\eta^1$-OSO photoisomeric species form within crystal structures of other [RuSO$_2$] complexes[44,45], this fleeting photochromism in **1** suggested that the temperature conditions of this experiment were almost suitable for $\eta^1$-OSO photoisomer formation.

We explored this suggestion by repeating the photocrystallography experiment on **1** at a slightly lower temperature, 90 K, using a new crystal to preclude any residual light-irradiation effects from the previous experiment. The crystal exhibited stable yellow-to-blue photochromism upon 505 nm light irradiation at 90 K. Accordingly, the light-induced crystal structure of **1** was determined at 90 K, uncovering a 100% photoconverted, ordered, $\eta^1$-OSO linkage isomer. ADPs for the Ru and S atoms and that of its pyridyl ring are systematically elongated along its main cationic axis (Fig. 1c, centre), which is indicative of a crystal structure manifesting close to a phase transition.

Nonetheless, the light-induced crystal structure of **1** at 90 K was well characterised, especially considering the clear identification of its $\eta^1$-OSO ligand, whose 100% photoconverted state means that its structural characterisation is not deterred by complications of positional crystallographic disorder that manifest in other [RuSO$_2$] complexes[36,43]: it is difficult to assign electron-density contributions to multiple atoms with partial occupancies (photoconversion fractions) that reside within the same geometric environ, within the periodic confines of a unit cell.

There are only two previously reported cases of 100% photoconversion in $\eta^1$-OSO photoisomeric crystal structures of [RuSO$_2$] complexes[42,44], which chemically differ from **1** via their *trans* ligands which are 3-bromopyridine or 3-iodopyridine. Thus, all known instances of 100% photoconverted $\eta^1$-OSO photoisomeric crystal structures of [RuSO$_2$] complexes involve heavy halogen (Br or I) substituted pyridyl *trans* ligands. This stands to reason because heavy-atom attachment to π-conjugated ligands is a commonly used molecular-design strategy for improving photostability in a compound; heavy atoms enhance the optical access of a compound to its higher-energy quantum states by strengthening spin-orbit coupling[54]. This heavy-atom attachment effect appears to dominate over other known factors that can affect photoconversion fractions; including the *trans* influence, given that the chloro analogues of 3- and 4-bromopyridine exhibit a similar level of *trans* influence; yet, their photocrystallographic findings are very different[36].

The formation of each $SO_2$-photoisomeric mode of **1** was accompanied by a large expansion in its crystallographic unit-cell length, *a*, by 0.42-0.55 Å and a very large contraction in unit-cell length, *b*, by over 1 Å (Table 1); *cf.* the largest structural perturbations in [RuSO$_2$] complexes seen until now only afforded unit-cell length changes of about 0.3 Å[44]. This high degree of unit-cell distortion in **1** indicates that the size and shape of the $SO_2$ reaction cavity, in which the $SO_2$ ligand has room for manoeuvre within its surrounding crystal lattice[30], is put under severe crystal-lattice strain owing to the formation of either photoisomer (see Supplementary Note 3). In turn, this large crystallographic change in **1** presumably accounts for the severe level of microscopic cracking that was observed in the crystal as a result of light irradiation, via optical microscopy (Fig. 2a). Indeed, changes in the crystallographic unit-cell parameters *a*, *b* and *c* of **1** run along the longest, middling and shortest sides of its crystals, respectively. The very large crystallographic contraction along *b*, with a lack of compensating contractions along other unit-cell axes, will essentially cause microscopic cracks to form in the crystal perpendicular to the *b*-axis i.e., cracks that run along the length of the main face of the crystal (*cf.* Fig. 2a).

### Optical-absorption spectral characteristics of the $\eta^1$-$SO_2$, $\eta^1$-OSO and $\eta^2$-(OS)O isomers in 1
A set of concerted single-crystal optical-absorption spectroscopy and microscopy experiments were systematically undertaken, using our custom experimental set up[41], to further characterise our single-crystal optical actuators.

Figure 2a displays the single-crystal optical-absorption spectra of **1** at light and temperature conditions that capture its $\eta^1$-$SO_2$ dark-state isomer, $\eta^1$-OSO photoisomer and $\eta^2$-(OS)O photoisomer, each in their 100% form. The ability to generate $\eta^1$-OSO and $\eta^2$-O(SO) photoisomers in **1** with 100% photoconversion, as well as distinguish the $\eta^1$-$SO_2$ dark-state isomer, provided a unique opportunity to isolate the optical-absorption spectral characteristics of all three $SO_2$ isomers within a single [RuSO$_2$] complex.

The spectrum for the $\eta^1$-$SO_2$ dark-state isomer of **1** exhibits one main absorption peak, within the wavelength range, *ca.* 425–625 nm, at full-width half-maximum (FWHM). This is consistent with the yellow hue of the crystal of **1** when probed in its dark state at 90 K (Fig. 2a, bottom left inset). This is also broadly characteristic of the spectral

**Table 1 | Crystallographic unit-cell parameters for the dark- and light-induced crystal structures of 1 that were determined at 90 K or 100 K**

|              | a (Å)      | b (Å)       | c (Å)       | α (°)      | β (°)      | γ (°)       |
|--------------|------------|-------------|-------------|------------|------------|-------------|
| **Dark, 100 K**  | 8.2544(3)  | 12.1811(4)  | 14.1063(5)  | 92.247(2)  | 97.270(2)  | 93.706(2)   |
| **Light, 90 K**  | 8.670(8)   | 11.157(10)  | 14.345(13)  | 91.804(15) | 93.284(15) | 93.939(14)  |
| **Light, 100 K** | 8.806(2)   | 11.139(3)   | 14.215(3)   | 92.024(3)  | 93.674(3)  | 93.950(3)   |

The shaded table cells highlight the large light-induced changes in unit-cell lengths.

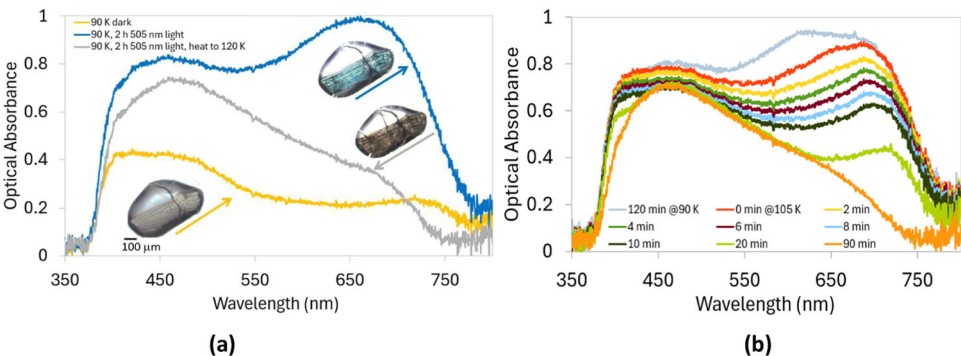

**(a)**                                             **(b)**

**Fig. 2 | Single-crystal optical-absorption spectra and optical-microscopy images of 1. a** Single-crystal optical-absorption spectra of **1** at light and temperature conditions that capture its η¹-SO₂ dark-state isomer, its 100% converted η¹-OSO photoisomer and its 100% converted η²-O(SO) photoisomer. (Insets) Optical-microscopy image of a crystal of **1** having been stimulated with 505 nm light for 2 h while held at 90 K (top right in (**a**)) and then heated to 120 K (middle right in (**a**)) before being heated to 295 K and re-cooled to 90 K (bottom left in (**a**)). Note that some residual microscopic crack lines persist in this re-cooled crystal, despite having thermally reverted to its 100% dark-state structure. **b** Single-crystal optical-absorption spectra of **1** acquired *t* minutes after the temperature of the crystal had reached 105 K, having been stimulated with 505 nm light for 2 h while held at 90 K before raising its temperature to 105 K.

profile for the dark-state η¹-SO₂ configuration in other [RuSO₂] complexes[38,40–45], the primary difference being that the dark-state spectrum of **1** absorbs light to a longer wavelength (*ca.* 625 nm at FWHM) unlike in [RuSO₂] complexes which contain 3-halopyridyl *trans* ligands, whose optical-absorption peak tails off by *ca.* 550 nm at FWHM[42,44,45].

The optical-absorption spectral profile for the η¹-OSO photoisomeric configuration of **1** spans the full panchromatic range of the visible spectrum, and features two broad absorption bands whose peaks are centred at *ca.* 450 nm and *ca.* 675 nm. The greater optical density of the longer-wavelength peak stands to reason given the very bright and distinctive turquoise-blue hue of the 505 nm light-induced crystal of **1** at 90 K that is observed via concerted optical microscopy (Fig. 2a, top right inset). These spectral characteristics of **1** are broadly similar to the panchromatic spectral signatures of the 100% photoconverted η¹-OSO species that were observed for the 3-bromopyridyl and 3-iodopyridyl-based [RuSO₂] complexes[42,44]. There is one notable exception in that the η¹-OSO photoisomer in **1** lacks the prominent peak that is centred at *ca.* 550-600 nm in the optical-absorption spectrum of its 3-halopyridyl analogues[42,44,45]. MLCT that is associated with nanooptomechanical transduction in these 3-halopyridyl [RuSO₂] complexes was deemed to cause such peaks. Given that **1** contains a 4-halopyridyl rather than 3-halopyridyl ligand, it cannot undergo that type of nanooptomechanical transduction, so it is bereft of that optical-absorption spectral peak.

The crystal of **1** in its η²-(OS)O photoisomeric state displays a dark yellow-grey hue (Fig. 2a, middle right inset). The corresponding optical-absorption spectrum is panchromatic across the visible range of light, albeit it displays one broad band whose peak is centred at *ca.* 460 nm, which tails off in the shorter-

wavelength spectral region. These spectral characteristics were generated by heating the η¹-OSO photoisomeric structure to 120 K; this was found to be an alternative route to form the 100% η²-(OS)O photoisomeric complex; as opposed to the aforementioned direct route of exposing 505 nm light to **1** while holding the crystal at 100 K.

## Thermal stability of η¹-OSO and η²-(OS)O photoisomers in 1

The η¹-OSO photoisomer is metastable at 90 K, as demonstrated by measuring the single-crystal optical-absorption spectrum of **1** as a function of time lapse from the time point where the crystal has been photostimulated for 2 h while held at 90 K (Supplementary Fig. S1). The thermal stability of the crystal of **1** at 90 K was also monitored by tracking its constancy of photochromism using the concerted optical microscopy part of the experimental setup. The turquoise-blue hue of the light-induced crystal of **1** at 90 K persisted for the full testing period of 1 h (Supplementary Movie 1).

The η¹-OSO photoisomeric structure of **1** was found to decay slowly to its η²-(OS)O photoisomeric form at 105 K, as indicated via the time-lapsed series of optical-absorption spectra that were measured over a 10 min period after the crystal had been photostimulated while held at 90 K for 2 h and raised to 105 K (Fig. 2b). Corresponding optical-microscopy images were captured for this same series of time points and beyond, up to a total time-lapse period of 90 min at 105 K (Supplementary Movie 2). These images show that the characteristic turquoise-blue hue of the crystal of **1** in its η¹-OSO photoisomeric state at 90 K (*cf* Fig. 2a inset, top right) switches to the yellow hue of its η²-(OS)O photoisomeric structure, before it reaches 105 K. This is consistent with the loss of the 615 nm peak in the optical-absorption spectrum of **1** that is

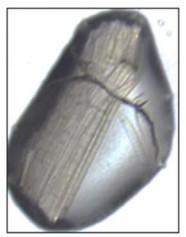 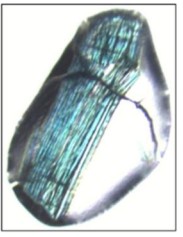 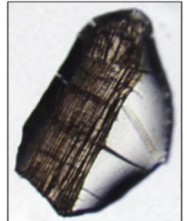 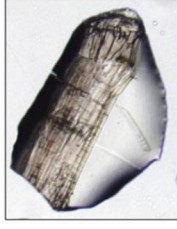 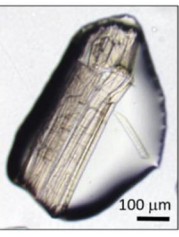

90 K (dark)　　90 K (light) **Heat to:** 110 K　　240 K　　260 K

**Fig. 3 | Optical-microscopy images that show photochromic changes in a single crystal of 1 as a function of increasing temperature.** Transmission optical-microscopy images of a single crystal of **1** in oil, first at 90 K (dark) whereupon the crystal is exposed to 2 h of 505 nm light, and is then heated progressively to 295 K in 8 temperature increments, all of which are shown in the Supplementary Fig. S3. The choice of 260 K as the last temperature in this figure (rather than room temperature) reflects the highest reliable temperature that the oil which holds the crystal in position is sufficiently viscous to maintain a solid film. The contrast of optical-microscopy imagery acquired at 295 K will be affected by the oil losing its viscosity at room temperature; nonetheless, the 295 K image is included in the Supplementary Fig. S3. A few black marks that extend into the clear area of the film in this series of images represent cracks in the film of frozen oil (rather than in the crystal) which form as a knock-on effect of macroscopic cracking in the adjoining crystal. Supplementary Movie 3 also shows the same trend in a different crystal, up to $T = 275$ K.

observed upon heating the crystal from 90 K and 105 K; once at 105 K, the spectra show that **1** takes tens of minutes to settle fully into its $\eta^2$-(OS)O configuration. This crystal of **1** was maintained in this configuration until it was heated further to *ca.* 200 K, whereupon it has fully reverted to its dark-state $\eta^1$-SO$_2$ isomeric form. Overall, this thermally-stimulated reverse-isomerisation sequence is broadly mirrored in other [RuSO$_2$] complexes[38,40–45].

**Light-induced microscopic and nanoscopic crystal strain in 1**
The optical-microscopy images also evidenced the aforementioned light-induced formation of severe microscopic cracking in crystals of **1**. Figure 3 and Supplementary Fig. S3 and Supplementary Movie 3 show how these cracks in crystals of **1** diminish progressively as its light-induced form at 90 K is heated to 275 K, thereby alleviating the causal crystal-lattice strain. While crystals of other materials have experienced light-induced crystal-lattice strain[11], this report presents a [RuSO$_2$] complex displaying microscopically visible crystal cracking from a light-induced process; moreover, its severity is very striking. We therefore decided to take a closer look at these cracks in crystals of **1**. Thereby, we employed in-situ light-induced atomic force microscopy (AFM) since the depths of the crystal fissures were too shallow to be observed by optical microscopy.

Figure 4 and Supplementary Movie 4 display a sequence of AFM images that were captured while 510 nm light was progressively shone onto the largest face of a crystal of **1** that was maintained below 90 K. The gradual build-up of light-induced crystal strain caused by $\eta^1$-OSO photoisomeric structure formation could therefore be monitored. Each AFM image conveys a $4 \times 8$ μm rendering of the crystal, as viewed looking down onto its main face, using a colour-scale to indicate the pattern and depth of its surface features that form as a function of light-exposure time; underneath each image is a cross-section of its nanoscopic depth profile that was taken from a cut along its $y = 0$ line, which helps to quantify the third dimension of the crack fissures caused by light-induced crystal strain. The $y = 0$ line was chosen as it allows one to observe structural features across all AFM images over a good dynamic range, while avoiding any human bias, e.g., a natural temptation to centre deliberately on the most intense features; a cross-section half-way down the image seems to be neutral.

Figure 4a shows the crystal surface of **1** in its dark state, which supports a subtle zig-zag shaped pattern that tracks down the image. This zig-zag pattern expands and contracts slightly as 510 nm light is initially applied (Supplementary Movie 4); this makes sense since we saw earlier that the SO$_2$-photoisomerisation process in **1** causes a large change in its crystallographic unit-cell dimensions. This movement is steadied by the growth of a vertical stripe that can be seen running

down the $x = -1.3$ μm axis after about 240 min of light exposure (Fig. 4e); this feature becomes clearer in subsequent images as the line progressively develops into a deep channel, *cf.* a light-induced nano-scopic crack forms within the crystal (Supplementary Movie 4). The zig-zag pattern on the right of the image starts to extend towards this channel as it becomes more stretched out and thus flatter, eventually affording two vertical stripes that run parallel to each other. At this point, these stripes are *ca.* 1.3 μm wide and their basins lie about 2 μm apart (Fig. 4i); the latter is consistent with the separation between light-induced cracks within the crystal of **1** that we observed via optical microscopy (Fig. 3 and Supplementary Movie 1). The respective depths of these two vertical stripes gradually homogenise, while a criss-cross feature forms in the bottom half of the image area. This process affords a wide channel that is *ca.* 250–300 nm deep and just over 4 μm in width (Fig. 4l).

The average root-mean-square (RMS) height across each entire AFM image (Supplementary Fig. S4a) increased monotonically as the crystal was progressively exposed to 510 nm light, becoming *ca.* 2.5 times higher than that of the crystal surface in its dark-state structural configuration once 100% photoconverted. This stands to reason because the optical-microscopy images suggest that just under one half of the crystal surface area is covered by dark striations while 100% photoisomerised (*cf.* Figure 3).

Figure 5 and Supplementary Movie 5 track the progressive closure of a crack fissure which demonstrates how the causal light-induced crystal strain in **1** is released by warming the crystal, whereby it was monitored as a function of increasing the temperature from 87.5 K to 295 K. The 4 μm wide channel closes up progressively, until it eventually appears as a small scar. The level of mechanical flux that the crystal undergoes through this process can be tracked via the RMS height which increases with temperature until 131 K (Supplementary Fig. S4b) where it reaches its maximum (RMS = 257 nm). **1** is well on its way to $\eta^2$-(OS)O to $\eta^1$-SO$_2$ reverse isomerisation by 130 K (Fig. 5g). Above this temperature, the closing motion comes in exclusively from left of the channel in order to cover it up, while the RMS height monotonically diminishes. The channel fully closes at 195 K (Fig. 5l); the RMS height stabilises to *ca.* 139 nm at this same temperature. These observations stand to reason since **1** is deemed to have recovered to its fully dark-state $\eta^1$-SO$_2$ configuration by *ca.* 195 K.

In conclusion, the light-induced microscopic striations in crystals of **1** form with nanoscopic depth, as witnessed by AFM. This single-crystal optical actuation is caused by SO$_2$ photoisomerisation, as evidenced by the high correlation that is observed between the thermally-induced crack depletion in **1** and the thermal stability of its SO$_2$

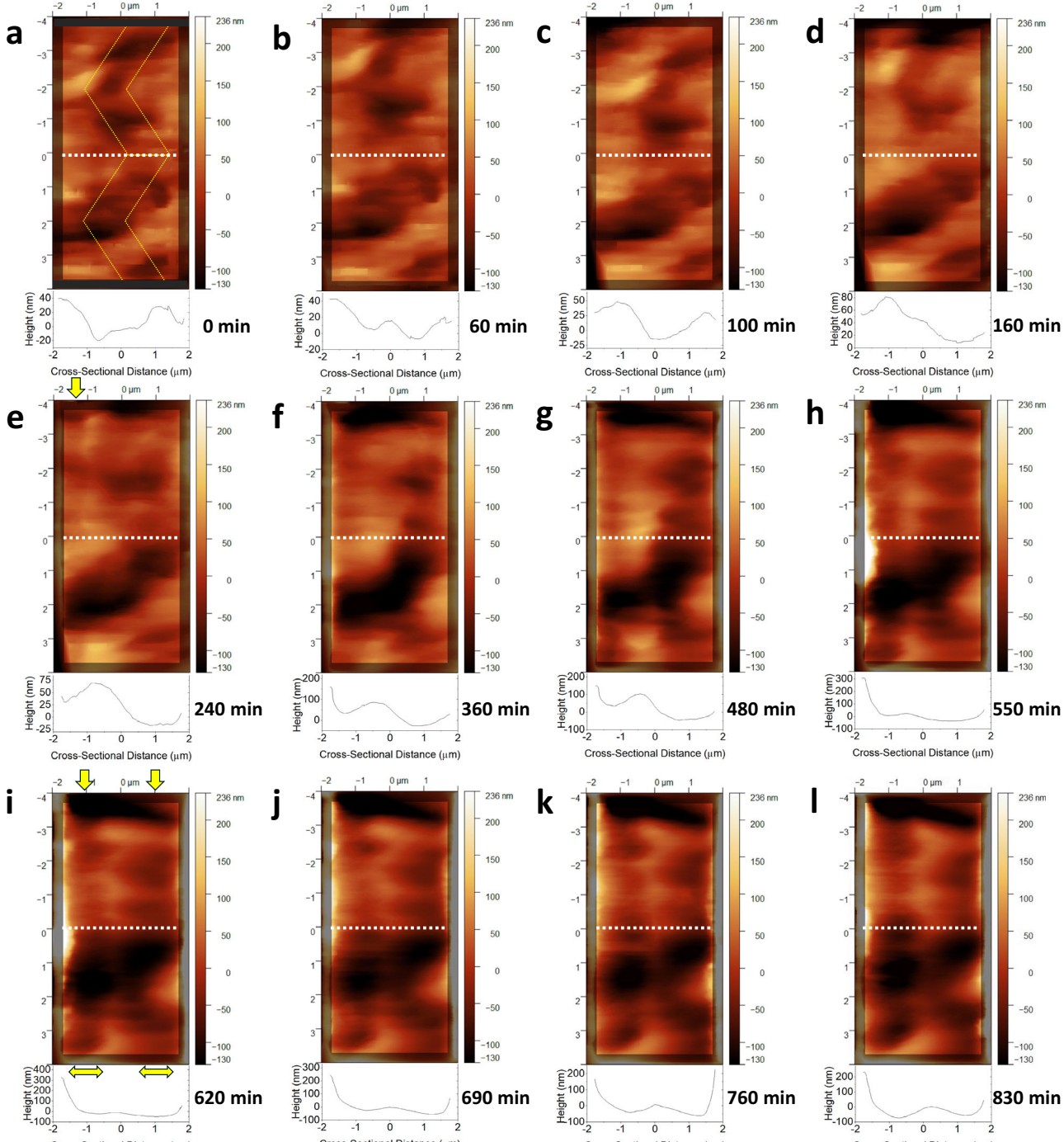

**Fig. 4 | The progressive light-induced formation of microscopic cracks across the surface of a single crystal of 1 whose channels are found to be nanoscopic in depth using atomic force microscopy.** A sequence of low-temperature atomic force microscopy (AFM) images that track the progressive in-situ light exposure of 510 nm light to the $4 \times 8$ µm surface area of a crystal of **1**. This crystal forms nanoscopic channels of cracks as it is exposed progressively to $t = 20$–830 min of 510 nm light at a glancing angle, as viewed via AFM; cross-sectional (depth) profiles are shown for the $y = 0$ (white dotted) line; the full range of crystal depth is displayed via the colour bars. Selected images are shown in Fig. 4 at $t =$ **a** 0, **b** 60, **c** 100, **d** 160 **e** 240 **f** 360 **g** 480 **h** 550 **i** 620 **j** 690 **k** 760 **l** 830 min. Yellow annotations guide the reader to a: **a** zig-zag feature **e** vertical stripe at −1.3 µm (arrow); **i** two vertical stripes (two arrows at top) centred at −1.0 and +1.0 that lie 2 µm apart and are 1.3 µm wide (two arrows at bottom). Supplementary Movie 4 shows the full series of AFM images for the crystal of **1** as it is progressively exposed to light.

photoisomers. The full recovery of its original macroscopic crystal form upon warming (Fig. 3) is striking and rare in nature. The unusually pure (100%) photoconversion of its η¹-SO₂ dark-state ligand into both of its η¹-OSO and η²-(OS)O photoisomeric forms, while **1** is held at 90 K and 100 K respectively, is technologically important. The associated light-activation processes were also found to be facile, requiring only a green light-emitting diode under liquid-nitrogen temperatures. It is

also practically attractive that **1** exhibits this light-induced phenomenon in its single-crystal form, i.e., a pure medium with long-range atomic order which is highly suited to photonic control. These factors combined suggest that our foundational discovery of this new material could lead ultimately to its development as a single-crystal optical actuator. This is important given the rising era of quantum technologies.

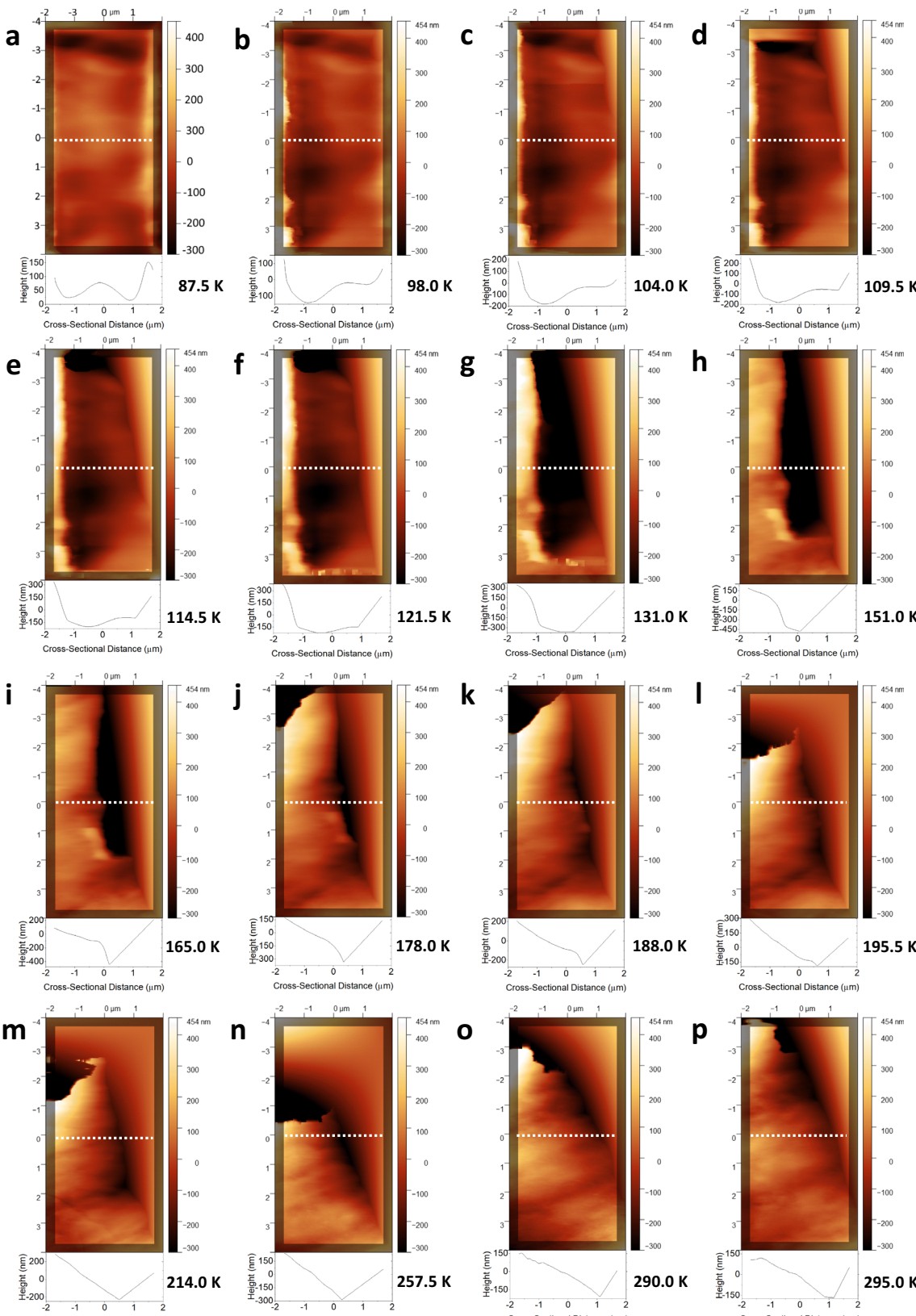

**Fig. 5 | The progressive closing up of nanoscopic channels of cracks in a single crystal of 1 as it is warmed up from 87.5 to 295 K, as witnessed via atomic force microscopy.** A sequence of low-temperature atomic force microscopy (AFM) images that track the progressive warming up of the light-exposed 4 × 8 μm surface area of a crystal of 1. The nanoscopic channels of cracks of the crystal close up as it is progressively warmed from 87.5 to 295 K, as viewed via AFM; cross-sectional (depth) profiles are shown for the y = 0 (white dotted) line; the full range of crystal depth is displayed via the colour bars. Selected images are shown in Fig. 5 at T = **a** 87.5 **b** 98.0 **c** 104.0 **d** 109.5 **e** 114.5 **f** 121.5 **g** 131.0 **h** 151.0 **i** 165.0 **j** 178.0 **k** 188.0 **l** 195.5 **m** 214.0 **n** 257.5 **o** 290.0 **p** 295.0 K. Supplementary Movie 5 shows the full series of AFM images for the crystal of **1** as it is progressively warmed up to 295 K.

## Methods

### Materials synthesis

The coordination complex, *trans*-[Ru(SO$_2$)(NH$_3$)$_4$(4-bromopyridine)] tosylate$_2$, (**1**) was synthesised from a precursor, *trans*-[Ru(SO$_2$)(NH$_3$)$_4$Cl]Cl, which was synthesised according to a literature procedure[55]. Thereby, 1.0 g (304 mol) of [Ru(NH$_3$)$_5$Cl]Cl was added to a solution of NaHSO$_3$ (40 ml, 0.3 M). The resulting reaction mixture was heated to 75 °C for 1 h under continuous SO$_2$ (g) sparging, and then cooled to room temperature while SO$_2$ sparging was maintained. The precipitating [Ru(NH$_3$)$_4$(H$_2$SO$_3$)$_2$] was collected by vacuum filtration and then added to 80 mL of a mixture of HCl and H$_2$O (50:50, v/v) and heated to its boiling point for 15 min. The resulting solution was filtered hot to remove any undissolved particles. The filtrate was cooled to room temperature overnight in the absence of direct light. The filtrate comprised single crystals of [Ru(NH$_3$)$_4$(SO$_2$)Cl]Cl which were washed with methanol before being dried at 60 °C for 1 h.

5 mg (16 μmol) of this precursor and 4-bromopyridine hydrochloride (11 mg, 57 μmol) were dissolved in a 0.1 M solution of Na$_2$CO$_3$ (1 mL) to afford a yellow solution. An aqueous 2 M (150 μL) solution of *p*-tosylic acid (>98% purity, Sigma Aldrich) was then added dropwise, which furnished yellow crystals within 2–4 h. Yellow rectangular crystals precipitated from solution; they were isolated via vacuum filtration and washed three times with methanol. The synthetic yield was not quantified since the single-crystal form of **1** was the desired product; although, the solution that produced **1** became translucent once the crystals had precipitated, indicating that the yield is high. The 3-D crystal structure of **1** was fully characterised in its dark and photoinduced states using standard single-crystal X-ray diffraction and photocrystallography[46–48], respectively.

### Materials characterisation

**Dark and photoinduced in-situ single-crystal X-ray diffraction of 1.**
The crystal structures of the dark- and light-induced states of **1** were determined using a single-crystal X-ray diffractometer at Argonne National Laboratory, IL, USA. The windows of this diffractometer were covered with aluminium foil to ensure a controlled light-induced experiment. A 0.45 × 0.15 × 0.100 mm$^3$ single crystal of **1** was mounted onto a three-circle Bruker diffractometer equipped with a monochromatic X-ray source (Mo Kα, $\lambda$ = 0.71073 Å), an Apex CCD detector and an Oxford Cryosystems open-flow N$_2$ cryostream which maintained the temperature of the crystal at 100 K. A series of data frames were acquired over multiple $\varphi$ and $\omega$ scans of crystal orientations, collected in 0.5° increments each with 60 s exposure time, while maintaining a 50 mm sample-to-detector distance. Data were reduced using SAINT v7.66 software, affording a total of 37892 or 46441 reflections for the dark- or light-induced data collections, respectively. Data for the dark-state crystal structure were first obtained. The crystal was then maintained at 100 K on the diffractometer and held static while 505 nm light was shone onto its thinnest face for 2 h, using a Thorlabs M505L3 light emitting diode (LED) whose head power output was 1000 mA constant current and 3.3 V forward voltage. The crystal was then illuminated for an additional 15 min at three rotated $\varphi$ orientations, 90°, 180° and 270° from its thinnest face, i.e., the crystal was photostimulated by 2 hr 45 min in total for the photocrystallography experiment. This light was switched off before acquiring data for the light-induced crystal structure. An analogous experiment was performed to determine the light-induced crystal structure of **1** at 90 K except that a fresh 0.700 × 0.100 × 0.050 mm$^3$ single-crystal was used. A total of 25100 reflections were realised from the light-induced data collection. Crystallographic Information Files (CIFs) provide further experimental details, as well as structure solution and refinement information; see Supplementary Note 4 and the Data Availability Statement. An overview of the specialised photocrystallography technique that is employed for these experiments is given elsewhere[46–48].

**Single-crystal optical-absorption spectroscopy and microscopy of 1.** A custom-built micro-spectroscopy system was used to record the absorption spectra of single crystals under a variety of light and temperature conditions. The system was built around an inverted microscope (Olympus: IX71) coupled to a 300 mm focal length spectrograph (Princeton Instruments: Acton Series 2300i) and 1320 × 100 channel CCD camera (Princeton Instruments: PIXIS 100BR). A 0.66 × 0.24 × 0.05 mm$^3$ crystal of **1** was mounted on a sapphire disk (9 mm dia., 0.5 mm thick) and fastened with a small amount of viscous perfluoroether oil. The mounted sample was then placed on the cold finger of an optical cryostat (Janis: ST-500-UC) attached to the microscope. The cold finger was drilled through, allowing optical absorption measurements to be made. The crystal was positioned such that only half of the active vertical channels of the CCD camera were used to image a portion of the crystal using a 5x, 0.13 NA objective (Olympus, NeoSPlan); the remaining half of the active detector channels imaged the sapphire substrate. The probe light for optical absorption measurements was provided by the microscope's 100 W tungsten-halogen lamp and 0.3 NA condenser optics. A visible band-pass filter (Schott: BG40) and OD 0.9 neutral density filter was placed between the lamp and the condenser to reduce the thermal load on the sample and cryostat.

To induce photoisomerisation, the crystal was held at 90 K or 100 K in the cryostat while being illuminated with 505 nm light from a ThorLabs M505F1 fibre optically coupled LED. The light from the LED was collimated and then coupled into the microscope through a side port, focusing it onto the back aperture of the objective, thus filling the field of view and evenly illuminating the entire crystal. The excitation power measured at the objective was typically 340 μW giving an estimated 15 μW/mm$^2$ illuminating the field of view.

Optical absorption spectra were recorded by imaging the crystal on the entrance slit (75 μm) of the spectrometer and dispersing the light, using a 150 line/mm grating, onto the detector. The image was positioned such that 10 rows of the detector were illuminated with light that passed through the crystal (I$_T$); whereas 10 rows of the detector directly above the crystal recorded light that passed through only the sapphire substrate (I$_0$). Absorption spectra were then calculated as $-\log_{10}$(I$_T$/I$_0$).

A more detailed description of this bespoke experimental setup and its operational pipeline, as provisioned to support photocrystallography, is given elsewhere[41].

**In-situ light-induced low-temperature atomic force microscopy of a single crystal of 1.** A 0.60 × 0.22 × 0.05 mm$^3$ crystal of **1** was mounted onto a sapphire disk (9 mm dia., 0.5 mm thick) using a two-component (one silver-filled) epoxy (EPO-TEK E4110-LV) for low-temperature affixation, which covered the underside of the crystal (Supplementary Fig. S5). The sample was loaded into the chamber of an Omicron UHV VT-AFM atomic force microscope that was equipped with a AppNano Al-coated silicon AFM tip and possessed sub-liquid nitrogen temperature capabilities which were administered by a Lakeshore 331 temperature controller. The sample was initially placed into a prep-chamber and subjected to a vacuum until it reached a pressure of 5.9 × 10$^{-10}$ mbar, at which point it was moved into the analysis chamber which operated with a pressure of 1 × 10$^{-11}$ mbar. An OptoEngine LLC 510 nm laser operating at 30 mW was positioned at the side of the AFM chamber exterior, and aligned such that it subtended a glancing angle with respect to the crystal. The reason for choosing a glancing angle was twofold: the angle of approach for the laser was thus restricted; the surface structure is premium for an AFM experiment, although we need to ensure that the entire crystal is bathed within the laser beam to avoid any unwanted structural gradient within the crystal. The AFM was operated by Nanonis electronics and control software. The AFM tip was aligned using an infra-red laser. The room lights housing the AFM instrument were then extinguished before the sample was cooled

to *ca.* 50 K and maintained below $T = 90$ K for the duration of the data acquisition of dark-state and light-induced AFM images. Technical AFM checks were conducted to verify that the tip was scanning true features (Supplementary Fig. S6). Having collected a dark-state image, two series of $4 \times 8$ μm rectangular AFM images of the crystal surface of **1** were then acquired in contact-mode. The first series of images pertained to an area on the crystal, which was progressively photostimulated by 510 nm light for 900 min while held at 80 K; this light exposure was set to be far longer than that of the X-ray diffraction or optical-absorption spectroscopy experiments owing to the glancing angle that the light subtends onto the crystal sample in the AFM experiment. Metastability checks were performed at the end of the total light exposure period, whereby two auxiliary AFM images were acquired below 90 K after the light had been removed for 30 min and 100 min; both images reproduced the features of the 900 min light exposed AFM image, thereby confirming metastability (Supplementary Fig. S7). The second series of scans were then repeatedly collected on the same area of the crystal, while progressively raising the temperature up to 295 K. All data were further processed and analysed using Gwyddion software[56].

## Data availability

Crystallographic Information Files (CIFs) and associated CheckCIF reports for the 100 K dark, 90 K and 100 K light-induced crystal structures generated in this study are provided; see Supplementary Note 4. A copy of these CIFs together with the X-ray crystallographic coordinates and structure factors for all crystal structures reported in this study have been deposited at the Cambridge Crystallographic Data Centre (CCDC), under deposition numbers 2365623 (dark, 100 K), 2365624 (light-induced, 90 K) 2365625 (light-induced, 100 K). These data can be obtained free of charge from The Cambridge Crystallographic Data Centre via www.ccdc.cam.ac.uk/data_request/cif. Supplementary Notes 1 and 2 also include the single-crystal optical absorption spectroscopy and microscopy and light-induced atomic force microscopy data as single images or a collated sequence of images (i.e., as movies). Additionally, all data are available from the corresponding author upon request.

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

## Acknowledgements

J.M.C. is grateful for the BASF/Royal Academy of Engineering Research Chair in Data-Driven Molecular Engineering of Functional Materials (RCSRF1819/7/10), which is partly supported by the Science and Technology Facilities Council via the ISIS Neutron and Muon Source. J.M.C. also thanks the 1851 Royal Commission of the Great Exhibition for the 2014 Fellowship in Design, hosted by Argonne National Laboratory where work done was supported by the U.S. Department of Energy (DOE) Office of Science, Office of Basic Energy Sciences, and used research resources of the Center for Nanoscale Materials, Office of Science User Facilities operated for the DOE Office of Science by Argonne National Laboratory, supported by the U.S. DOE, all under contract no. DE-AC02-06CH11357. J.d.J.V.-G. acknowledges the National Council of Science and Technology of Mexico (CONACyT) and the Cambridge Trust for a PhD Scholarship (217553).

## Author contributions

J.M.C. conceived and designed the project. J.M.C. performed all of the photocrystallography, in-situ light-induced atomic force microscopy (AFM), optical microscopy and absorption spectroscopy work, with experimental assistance from J.R.G. in the in-situ light-induced AFM experiment, and D.J.G. in setting up the optical microscopy and absorption spectroscopy apparatus, and the laser set up for the AFM experiment. J.M.C. carried out the data analysis. J.M.C. was the Ph.D. supervisor of J.d.J.V.-G who synthesised the material. J.M.C. drafted the manuscript. All authors provided input and agreed on the final manuscript.

## Competing interests

The authors declare no competing interests.
