## [Transparent Peer Review file · Nature Communications]

Ternary molecular switching in a single-crystal optical actuator with correlated crystal strain

Corresponding Author: Professor Jacqueline Cole

Version 0:

Reviewer comments:

Reviewer #1

(Remarks to the Author)

In the present work, the authors reported a new [RuSO₂] complex that produces a 100% photoconverted η^1 -oso isomeric crystal structure at 90 K, which completely transformed to the η^2 -(OS)O photoisomers when heated to 100 K, and a complete recovery of the dark state η^1 -SO₂ structure when the crystals were heated to room temperature. It was demonstrated that three different 100% converted SO₂-linked isomers exist in [RuSO₂] complexes. It revealed the possibility of creating high-purity ternary photonic features in single-crystal optical actuators. I recommend its acceptance before the authors properly responded the following concerns.

1. Did the authors attempt to change the conditions of the heating reaction so that the η^2 -(OS)O photoisomeric species to gradually return back to η^1 -OSO photoisomeric then to η^1 -SO₂? For example, the wavelengths of light.
2. In the Introduction section, when mentioning various applications of single-crystal optical actuators, some other literatures should be cited: Chem. Soc. Rev., 2024, 53, 5227; Angew. Chem. Int. Ed., 2024, 63, e202409472; Angew. Chem. Int. Ed., 2023, 62, e202302429; Angew. Chem. Int. Ed., 2023, 62, e202306048.
3. It is suggested that complementary kinetic studies of isomeric transformations at each step would be beneficial for understanding the transformation process.

Reviewer #2

(Remarks to the Author)

Cole et al. reported a new complex, trans-[Ru(SO₂)(NH₃)₄(4-bromopyridine)]tosylate₂ (1-SO₂), that shows optical switching as well as thermo/photoconversion properties with a combined change in temperature and light. Authors has successfully able to isolate the metastable products with 100 % conversion. Further, the reversible nature of isomerisation process seems interesting. The thermally reversible micro- and nanoscopic crystal strain, are well characterized with the help of optical microscopy and in-situ light-induced AMF studies. The following point need to address before publication in the journal:

1. In the above isomerisation process from dark 100K to light 90K/100K authors have used two stimuli (heat and light) simultaneously and in the reverse process (light 100K to dark 90K) only heat. However, it is not clear whether the isomeric crystal structures of [RuSO₂] complexes are purely photo-induced or thermally induced. Explanation or additional experimental results are needed.
2. In Fig S2 (supporting information), the optical image of the crystal at 260 K looks really crack free with clear crystal surface. However, no single crystal data is available at RT. As authors have not performed any thermal analysis, it is remained unclear how the crystal behaves at room temperature.
3. The isomerisation occurs with a clear bond breaking/making around the SO₂ with respect to metal ion. Do the space around the SO₂ group is changing and impacting any lattice rearrangement within the crystal? The surface nanoscopic crystal strain can be easily correlated to the inner structural changes.
4. Is there any change in intermolecular interaction during the photo isomerization (1-SO₂ to 1-OSO to 2-(OS)O isomers. How does these intermolecular interactions contribute to the stability to system? A detailed theoretical study may be added.
5. Only two meta stable isomers (light 90K, light 100k) are successfully isolated by SXCRD analysis. However, SCXRD may not able to isolate low stable meta-states. Additional experiments or a detailed theoretical study may add more insight of it. A variable temperate PXRD may be useful also.
6. The observed surface nanoscopic crystal strain is purely qualitative and has not quantified. A detailed nanoindentation on

the crystal surface may help in this regard and can be added.

Minor comments

1. Improve the introduction section with a broad (types) isomerization process exists in inorganic systems where meta stable exists and have successfully isolated.
2. The resolution of ORTEP in figure 1 looks very poor. Change to a better one.

Reviewer #3

(Remarks to the Author)

The article 'Ternary molecular switching in a single-crystal optical actuator with correlated crystal strain' is dedicated to one of the popular modern topics of mechanical response to chemical transformations in solids (about 40 years should have passed since the first quite deep studies of photomechanical and chemomechanical phenomena, so that the community began to show massive interest in these issues, at least at the phenomenological level). This paper is precisely an example of research useful for the development of this direction. I consider the attention to the temperature dependence of the Ru-SO₂ complex photoisomerisation pathways, as well as the attention to the factors of changes in structural parameters caused by the transformation, to be a particularly valuable feature (and result) of the study. Therefore, I believe that the paper can be published in Nature Communication. Nevertheless, some changes are needed - there are both formal comments on the text and conceptual questions, the answers to which would improve the article.

1. There seems to be a number of inconsistent designations for the bidentate type of Ru-(OS)O coordination in the text. In different parts of the text there are designations of the form: h1-(OS)O, h2-O(OS) and h2-(OS)O.
2. There is some disagreement about the light wavelength used in the AFM experiment. The description of the technique says 510 nm, but the caption to Figure 4 says 505 nm.
3. The description of the Figure 4 in the last paragraph on page 8 (starting with 'Figure 4a shows the crystal surface...' and ending with '...just over 4 μ m in width (Figure 4d).') sounds almost completely incomprehensible. Perhaps these features in the figures mentioned there need to be labelled somehow?
 - a. Where is the zig-zag pattern in Fig. 4a?
 - b. Where is the vertical line at $x=-1.6$ μ m in Fig 4b?
 - c. Where are the two vertical lines in 4c?
 - d. What was the point of choosing the section line at $y=0$? It seems to run along an area of low information, while the lower half of the image shows the appearance of deep crack-like areas.
4. The use of the terms microscopic and nanoscopic strains seems inappropriate. What is meant in the text are defects (cracks or pores) of microscopic and nanoscopic scales, so they should be called that. The formations seen here on crystal surfaces in the optical microscope and AFM are defects (such as cracks or pores) due to deformation caused by the isomerisation. Instead, the authors refer to the defects as strains, which is incorrect.
5. Fracture is visible on the optical microscopy images, even on the surface of the original crystals. This may mean that the crystals were fractured under irradiation by scattered daylight, even before they were irradiated in the experiment. This is not surprising since the absorption spectrum occupies the entire visible part of the spectrum and the transformation causes a fairly significant compression by $\sim 10\%$ along the b crystal axis. If this is true, one could say that all the photomechanical phenomena observed here are related to the opening (in the isomerised states) or collapse (in the stable isomer state) of the crack cavities. This raises two questions:
 - a. Did the authors not try to minimise the illumination of the crystals during their growth and preliminary operations on them?
 - b. Did the authors not observe fracture-free crystals of smaller size, hypothetically less than 5-10 μ m thick? If so, such crystals could demonstrate reversible bending or even twisting into a spiral (since the structure has triclinic symmetry) instead of fracture.
6. What are the directions of the crystal axes a, b and c in crystals? This information should be available after single-crystal XRD. It could help in interpreting the observed effects. From the orientation of the cracks on the surface visible in Figure 3, one can hypothesize that b axis is almost perpendicular to the long axis of the crystals - structure compression by $\sim 10\%$ along b axis should cause fracture in the surface layers, leading to the formation and opening of cracks perpendicular to b.
7. The cracks on the crystal surface are due to the inhomogeneity of photoisomerisation, whose scale is determined by the light absorption depth in this substance. Since the visible fracture scale is ca 10 μ m, one can assume that the absorption depth is also close to this scale. The depth of absorption is a very important property that determines the type of the mechanical response, which can be understood from [7] on the reference list, as well as from other works by these authors. If the depth is on the order of the crystal thickness, the crystal won't crack, but will bend or twist into a spiral, depending on the symmetry. If the crystal is significantly thicker (e.g. by an order of magnitude or more), the transformation will typically lead to fracture, since the strain inhomogeneity is too great for the crystal to simply bend without fracture. Moreover, if the thickness of the crystal is much greater than the absorption depth, it may take a long time to approach 100% transformation of the entire crystal volume (irradiation intensity and transformation rate decrease exponentially in deep regions of the crystal). This raises a number of other questions:
 - a. Did the authors estimate the absorption depth (from direct measurements at their facility or from absorbance obtained from UV-Vis spectra of the substance in solutions)? How is it related to the absorption depth?
 - b. What exactly is meant by "100% conversion" in the text: near 100% conversion of the entire volume of the crystals (~ 100 μ m thick), or 100% conversion in the surface layer of the crystals? If, as suggested above, the absorption depth is ~ 10 μ m, then the typical photon flux of the 3 W LED used in the experiments here is almost sufficient to achieve almost complete conversion to a depth of 100 μ m in 2 hours (assuming a high quantum yield of the isomerisation). However, if the absorption is much higher or the quantum yield is low, complete transformation of the entire volume may not be achieved.

Version 1:

Reviewer comments:

Reviewer #1

(Remarks to the Author)

The authors appropriately replied the questions raised by the three reviewers and made proper revisions in their revised MS. The results are solid and this work will attract a wide readership of researchers involved in photochemistry, photo-driven devices and soft robotics, etc.. I believe this revised manuscript could be accepted and published in this journal without additional corrections.

Reviewer #2

(Remarks to the Author)

The authors answered most of the questions raised by the three reviewers. The revised version of the manuscript has been significantly improved after incorporating the reviewers feedback. I could recommend its acceptance in the present form.

Reviewer #3

(Remarks to the Author)

While I am in disagreement with a number of the authors' conclusions and assertions (which I will discuss in further detail below), I would like to express my gratitude to them for their meticulous consideration of all the issues raised. Furthermore, I would like to note that the amendments made to the text make it clear and suitable for publication in Nature Communication. As a point for discussion, which does not affect my decision on the possibility of publishing this manuscript, I would like to highlight facts that may have led to a potential misunderstanding of the nature of some of the results.

Firstly, the authors' assertion that the absorption is not linear and does not obey the Beer-Lambert law (as well as similar assertions in the reference Corbett, D.; Warner, M. Linear and Nonlinear Photoinduced Deformations of Cantilevers) is erroneous. A potential misunderstanding of the Beer-Lambert law itself is indicated in Phys. Rev. Lett. 2007, 99, 174302, DOI: 10.1103/physrevlett.99.174302. In the cases under consideration, it would appear that the authors are suggesting that due to the alteration in the chemical composition of the phase (the occupation of positions by different isomers) caused by isomerisation, the extinction of the substance undergoes a change and, in general, becomes spatially non-uniform. This, in turn, results in the law of change in the intensity of healing with depth becoming non-exponential. Nevertheless, this does not imply that the principle of linear absorption, which forms the basis of the Lambert-Beer law, is no longer fulfilled. Its differential form remains linear in intensity: $dI/dx = -a(C) \cdot I$. The deviation from the exponential form $I = I_0 \cdot \exp(-a_0 \cdot x)$ arises due to the non-uniformity of the absorption coefficient $a(C)$ (if the concentration of isomers $C(x)$ is non-uniform, then $a(C(x)) = \text{const}$), rather than due to the non-linearity of the absorption process itself. Nonlinear absorption is a completely separate phenomenon, characteristic of nonlinearly absorbing substances (processes such as two-photon absorption, for example, leading to quadratic contributions to the absorption in intensity), and arising, in any case, at very large absorption coefficients and high intensities.

Secondly, optical absorption values in Fig. 2, which do not exceed approximately 0.8 for 505 nm radiation, provide clear evidence that the crystal thickness does not exceed $1.6 = -\ln(0.2)$ times the characteristic absorption depth. It is therefore unlikely that the deformations observed under such conditions will result in significant stresses within the crystal, nor will they be capable of causing fracture. Instead, the crystal will experience macroscopic deformation that is almost entirely consistent with the inhomogeneous distribution of isomers along the thickness, arising at an intermediate stage of transformation and residual stresses. For further verification of this claim, the reader is directed to the mentioned work of Corbett and Warner (2007) and mostly to ref. [10] of this work (M. Warner and L. Mahadevan, Phys. Rev. Lett. 92, 13430). However, the absorption of red light at ~650 nm is markedly high, which could potentially lead to significant inhomogeneity in the transformation at a later stage, capable of inducing high tensile stresses in the b direction and causing fracture. Related to this was my question about preventing the crystals from being illuminated before the experiment. My assumption, consistent with the results observed in the work, remains that authors always worked with defective crystals and simply observed crack opening or closing under the action of the resulting deformations.

Version 2:

Reviewer comments:

Reviewer #3

(Remarks to the Author)

I would like to express my gratitude to the authors for their responses and would like to recommend that this manuscript be published without further revision.

Dear Editor,

Many thanks for your work on our paper. We have now revised the manuscript according to the feedback from the three reviewers and we herewith respond point-by-point to all queries. Our responses are given in blue font, which have been inserted below after each point made by each reviewer whose comments are reproduced in black font. Associated revisions to the manuscript are highlighted using 'track changes' in a marked manuscript that we submit together with a clean version for production. The paper has certainly been improved from responding to this review process. Accordingly, we would like to thank the reviewers for all comments and advice.

Best wishes,

Jacqui Cole (on behalf of all authors)

REVIEWER COMMENTS

Reviewer #1 (Remarks to the Author):

In the present work, the authors reported a new [RuSO₂] complex that produces a 100% photoconverted η^1 -oso isomeric crystal structure at 90 K, which completely transformed to the η^2 -(OS)O photoisomers when heated to 100 K, and a complete recovery of the dark state η^1 -SO₂ structure when the crystals were heated to room temperature. It was demonstrated that three different 100% converted SO₂-linked isomers exist in [RuSO₂] complexes. It revealed the possibility of creating high-purity ternary photonic features in single-crystal optical actuators. I recommend its acceptance before the authors properly responded the following concerns.

Thank you for this positive recommendation of our work for *Nature Communications*.

1. Did the authors attempt to change the conditions of the heating reaction so that the η^2 -(OS)O photoisomeric species to gradually return back to η^1 -OSO photoisomeric then to η^1 -SO₂? For example, the wavelengths of light.

The effect of temperature was explored – in fact, this is how this low-energy η^1 -OSO species was initially discovered – Cole was collecting X-ray diffraction data on a crystal of this complex at 100 K and the photochromic change that was initially observed upon exposure to 505 nm light was found to be unstable. It is known from studies on other complexes in this series of [Ru(SO₂)(NH₃)₄X]Y compounds that the η^1 -OSO species is responsible for photochromism; yet, this instability at 100 K suggests that this species is unstable at this temperature. Cole naturally sought to make the photochromic change stable by cooling the temperature down to the base limit of the open-flow N₂ cryostream on the X-ray diffractometer. 90 K was a reliable temperature that avoided any technical icing problems, and exposure of 505 nm light to the crystal at 90 K afforded stable photochromism. Accordingly, the η^1 -OSO crystal structure was obtained at this temperature, and its 100% photoconversion level was found upon crystal structure refinement. The analogous X-ray diffraction experiment at 100 K yielded the 100% converted η^2 -(OS)O species, as has already been discussed in the paper.

The single-crystal optical absorption microscopy and spectroscopy setup had a similar baseline temperature constraint of 90 K, and analogous explorations of heating up the crystal to 105 K (Movie S1 versus Movie S2) and progressively from 90 K to 275 K (Movie S3) showed the same photochromic changes in the microscopy; while the spectroscopy of the crystal when exposed

to 2h light at 90 K and then heating it to 105 K and waiting for a period of time, showed signatures of thermal decay of the η^1 -OSO state to the η^2 -(OS)O species from 90 K to 105 K (*cf.* Figure 2b in the manuscript). Another spectroscopy experiment was undertaken that exposed the crystal to 2h light at 90 K and then heated it to 120 K, 160 K, 205 K. See the plot immediately below. This shows optical absorption signatures that depict the η^1 -OSO state stability at 90 K, the η^2 -(OS)O species at 120 decaying a bit by 160 K and fully reverting to the dark state by 205 K (*cf.* its comparable signature with that of its original dark state). **We have now added the results of this experiment to section A1 of the Supplementary Information to clarify this matter (see the new Figure S2).** NB: labelled arrows below are just for reviewers, not the SI.

Certainly, it is clear from this work, and from previous metrological work on other complexes within this family of compounds (references 27-45 as cited within the manuscript), that the η^1 -OSO species is thermally less stable than the η^2 -(OS)O species of this complex.

The wavelength was not varied for two reasons: (i) a shorter wavelength (higher energy) would be needed to have any chance of changing the order of stability between η^1 -OSO and η^2 -(OS)O species, given that the former is less thermally stable than the latter at 505 nm illumination. Even then, this is unlikely to yield any change because the optical absorption profiles for η^1 -OSO and η^2 -(OS)O species shown in Figure 2(a) reveal that the optical density of the former is consistently greater than that of the η^2 -(OS)O species at all wavelengths shorter than the 505 nm light used in this experiment. (ii) there is a substantial body of work published about this series of complexes (references 27-45 as cited within the manuscript) and there has never been any hint that the η^1 -OSO species becomes more stable than the η^2 -(OS)O species in a $[\text{Ru}(\text{SO}_2)(\text{NH}_3)_4\text{X}]\text{Y}$ complex; and yet, there is plentiful and consistent evidence of the reverse, *i.e.*, that η^1 -OSO is less stable than the η^2 -(OS)O species. While the present study does reveal that the stability of these two species in the subject compound are closer than have been seen in previous studies, the aforementioned rationale posed within (i) leaves little room for doubt that this order of stability will change when photoexcited at a shorter wavelength.

In contrast, what is known is that the η^2 -(OS)O species in this series of complexes will revert to the η^1 -SO₂ dark-state species upon application of red light or heating; see, for example, work cited as 27 or 40 in this manuscript.

2. In the Introduction section, when mentioning various applications of single-crystal optical actuators, some other literatures should be cited: Chem. Soc. Rev., 2024, 53, 5227; Angew. Chem. Int. Ed., 2024, 63, e202409472; Angew. Chem. Int. Ed., 2023, 62, e202302429; Angew. Chem. Int. Ed., 2023, 62, e202306048.

Thank you for providing these references to the wider application area of microrobotics; we have now added these citations to the introduction section of our revised manuscript.

3. It is suggested that complementary kinetic studies of isomeric transformations at each step would be beneficial for understanding the transformation process.

Thanks for this suggestion. Kinetic studies of isomeric transformations of single-crystal optical actuators might normally be done by single-crystal X-ray diffraction so that one can quantify the changes in SO₂ population as a function of time. However, they are not practical to undertake via single-crystal X-ray diffraction in this study owing to the macroscopic fracture that typifies the optical actuation in this material: even though the fracture recovers by warming up to room temperature, the macroscopic fracture observed in these single crystals makes it difficult to realise high quality crystal-structure determinations at the level by which one could observe subtle changes to the crystallographic occupancy factor of SO₂ atoms over the dynamic timeframe of a kinetic study. One can gain an appreciation of this challenge by taking a look at certain structural characteristics of the light-induced crystal structures that we have already obtained: key indicators are the various R-factors associated with each crystal structure determination in the Crystallographic Information Files that we have provided:

cf. indicators of the internal consistency of the data: R(equiv) and R(av sig/I) which are 5.6% and 5.4% for the 100 K dark-state (*i.e.* values that one could expect for a regular crystallographic dataset); compared with the higher 15.6% and 12.5% for the 100 K light-induced structure (the marked lattice distortion caused by forming the η^2 -(OS)O species); and even higher 33% and 49% for the 90 K light-induced dataset (given the η^1 -OSO species causes a major level of macroscopic fracture as well as marked lattice distortion).

cf. indicators of the quality of the crystal-structure refinement: R(all) and R(I>2sig(I)) which are: 7.5% and 5.2% for the 100 K dark-state (*i.e.* values that one would expect for a crystal structure) 15.4% and 8.2% for the 100 K light-induced crystal structure (showing an issue with weak data, which reflects the marked lattice distortion caused by the formation of the η^2 -(OS)O species) 30% and 9.7% for the 90 K light-induced crystal structure (showing a worse issue with weak data, reflecting the major level of macroscopic fracture caused by forming the η^1 -OSO species).

These qualifiers also evidence just how challenging the current photo-crystallographic study has been from an experimental data acquisition and data-analysis perspective.

While this precludes a quantitative approach to kinetic studies via diffraction, we considered what semi-quantitative insights may be possible from single-crystal optical absorption spectroscopy where one can obtain data on a particular part of the crystal (*i.e.* using it as a local rather than bulk materials-characterisation tool).

Considering each isomeric state in turn:

The stability of the η^1 -OSO species (light-induced state at 90 K):

Figure S1 in the Supplementary Information shows that the 90 K light-induced state (the η^1 -OSO species) is stable for at least 30 min and its associated microscopy shows that it is stable for at least 60 min (see Movie S1). Furthermore, we also know empirically that this state is stable for at least 36 h as this is the timeframe in which the photocrystallographic data were acquired from which a crystal-structure refinement was obtained with 100% η^1 -OSO species).

Isomeric transformation of the η^1 -OSO species to the η^2 -(OS)O species (kinetics at 105 K):

Figure 2(b) of the manuscript shows how the η^1 -OSO species gradually decays into the η^2 -(OS)O species once the crystal has been warmed up from 90 K to 105 K with the temperature being then held constant. The series of optical absorption spectral profiles in Figure 2(b) evidence a gradually decaying peak centred at ca. 685 nm (λ_{max} , red); this peak is therefore characteristic of the η^1 -OSO species. One can therefore use the observed changes in the optical absorbance (OA) of this peak to calculate a half-life for the η^1 -OSO at 105 K.

cf. first, calculate the ratio of OA for the crystal at λ_{max} of the red peak (ca. 685 nm) for 105 K at $t = 0$ (i.e. having just reached 105 K) relative to that of OA at $t = n$:

t /min	OA	Ratio [OA (λ_{max} , red peak), $t = 0$ / OA (λ_{max} , red peak), $t = n$]
0	0.876	
2	0.814	0.92922374
4	0.768	0.87671233
6	0.729	0.83219178
8	0.671	0.76598174
10	0.623	0.71118721
20	0.432	0.49315068

This shows that it takes 20 min for this OA characteristic of the η^1 -OSO species to become half of its OA value when the η^1 -OSO species can be considered to be at its starting point of decay from ~100 % (judging from the comparable optical absorption spectral profiles of the $t = 0$ min spectrum at 105 K (Figure 2(b)) and the light-induced spectrum at 90 K (Figure 2(a))).

Thus, the half-life, $t_{1/2}$, of the η^1 -OSO species at 105 K is approximately 20 minutes.

i.e. the η^1 -OSO species thermally decays into η^2 -(OS)O species at 105 K with $t_{1/2} = 20$ min.

One should bear in mind that these kinetic insights are limited owing to the use of calculations that focus only on the red peak in these spectra. However, it provides a semi-quantitative benchmark to readers about the likely temperature conditions and rates at which these photo-linkage isomers can form or decay.

The η^1 -SO₂ species (dark-state species):

Some basic insights about the formation of the η^1 -SO₂ species can be obtained from the new Figure S2 in the Supplementary Information (this Figure was created as per our response to the first point made by this reviewer). This Figure S2 provides a thermal stability plot of the three isomeric states. The η^2 -(OS)O species dominates the spectral profiles at 120 K and 160 K (a small residual 'red peak' characteristic of the η^1 -OSO species remains at 120 K but not 160 K).

The η^2 -(OS)O species appears to have fully reverted back to the η^1 -SO₂ dark-state species by 205 K, judging from the highly comparable profiles of the spectra at 90 K (dark) and 205 K (light induced, then warmed up to 205 K).

The crystal was warmed up from 90 K to 205 K at *ca.* 3 K per minute (stopping for just 2 min to acquire data at certain temperatures *en route*). Given that the 160 K spectrum is dominated by the η^2 -(OS)O species and the 205 K is essentially indistinguishable from a dark-state spectrum, one can at least benchmark the kinetics of the η^2 -(OS)O to η^1 -SO₂ reverse isomerisation using the calculation:

Temperature change: 205 – 160 K = 45 K

Rate of temperature change is 3 K per minute

Thus, it takes $45/3 = 15$ min for the η^2 -(OS)O species to revert to the η^1 -SO₂ species, when the crystal is warmed up from 160 K to 205 K at 3 K / min.

This is in line with what is known from previous studies about the thermal stability of the η^1 -SO₂ dark-state species in this series of complexes, whereby the η^1 -SO₂ dark-state species tends to form at *ca.* 200 K where the ligand *trans* to SO₂ is based on a pyridyl ligand.

Thus, the thermal stability of the η^1 -SO₂ dark-state species in the subject compound can be considered to be typical for a compound in this series of complexes.

This conclusion appears to be fit for purpose given that the photo-isomerised species are the primary focus of this study, and a number of previous studies have characterised the thermal stability of the η^1 -SO₂ dark-state species.

Overall, we have explained the level of semi-qualitative insights that we can obtain from optical absorption spectroscopy given that the macroscopic fracture precludes gaining quantitative information from X-ray diffraction. We already presented a dedicated section in the originally submitted paper about the thermal stability of the two linkage photoisomers of the subject compound, and the above detailed analysis essentially mirrors the salient conclusions of that section of the paper which is encouraging. Given this and that the above extra insights are only semi-qualitative, we suggest that we do not include this detailed information in the paper or its Supplementary Information. Yet, we hope that this explanation is helpful to guide the reviewer. Besides, the data from which we have derived this very detailed information already exists in the paper or Supplementary Information in its revised form such that the interested reader could regenerate it if required. Many thanks for prompting us to consider this point in such detail.

Reviewer #2 (Remarks to the Author):

Cole et al. reported a new complex, *trans*-[Ru(SO₂)(NH₃)₄(4-bromopyridine)]tosylate₂ (η^1 -SO₂), that shows optical switching as well as thermo/photoconversion properties with a combined change in temperature and light. Authors has successfully able to isolate the metastable products with 100 % conversion. Further, the reversible nature of isomerisation process seems interesting. The thermally reversible micro- and nanoscopic crystal strain, are well characterized with the help of optical microscopy and in-situ light-induced AMF studies.

Thank you for this positive review of our work.

The following point need to address before publication in the journal:

1. In the above isomerisation process from dark 100K to light 90K/100K authors have used two stimuli (heat and light) simultaneously and in the reverse process (light 100K to dark 90K) only heat. However, it is not clear whether the isomeric crystal structures of [RuSO₂] complexes are purely photo-induced or thermally induced. Explanation or additional experimental results are needed.

Thanks for this query, which shows that we need to make a little clearer in a few places of the paper that heat and light are not changed simultaneously. In fact, the crystal is photoisomerised by exposing it to light *while it is maintained at a given (fixed) low temperature*. We've checked through the manuscript to find places where this is mentioned: it appears to be clear in most places but there are a few instances where we could add a few words to make it clearer. Accordingly, we have clarified the text in the following instances where we can see that it could be clearer (highlights, in bold italics, show these additions or modifications to the original text):

In the introduction section:

“Many variations of X and Y can support one or both η^1 -OSO or η^2 -(OS)O photoisomeric configurations within the single-crystal form, whose **light-induced** structure is metastable when kept suitably cooled (typically to 100 K) and which thermally reverts to its η^1 -SO₂ dark-state crystal structure.” (“light-induced” is new text to make this explicitly clear).

The caption of Figure 1(b):

“...caused by 100% SO₂ linkage photoisomerisation that occurs when **the crystal is exposed** to 505 nm light **while maintained** at low-temperature: the η^1 -OSO or η^2 -(OS)O photoisomer forms at 90 K or 100 K, respectively. The latter photoisomer can also be generated by warming up the former by 10 K.”

The caption of Figure 2 (in two places):

“Optical-microscopy image of a crystal of **1** having been stimulated with 505 nm light for 2 h **while held** at 90 K” and “having been stimulated with 505 nm light for 2 h **while held** at 90 K before raising its temperature to 105 K.”

The last sentence of the section entitled “Optical-absorption spectral characteristics ...”

“as opposed to the aforementioned direct route of exposing 505 nm light to **1 while holding the crystal** at 100 K.”

The first sentence of the section about “Thermal stability...”

“spectrum of **1** as a function of time lapse from the time point where the crystal has been photo-stimulated for 2 h **while held** at 90 K”

The second paragraph of the section about “Thermal stability...”

“after the crystal had been photo-stimulated **while held** at 90 K for 2 h”

In the conclusions section:

“The unusually pure (100%) photoconversion of its η^1 -SO₂ dark-state ligand into both of its η^1 -OSO and η^2 -(OS)O photoisomeric forms, **while 1 is held at 90 K and 100 K respectively**, is also unprecedented.”

In the “Methods – Single-crystal optical-absorption spectroscopy ...” section:

“To induce photoisomerisation, the crystal was *held at 90 K or 100 K in the cryostat while being* illuminated with”

In the “Methods – In-situ light-induced low-temperature atomic force microscopy...” section:

“which was progressively photo-stimulated by 510 nm light for 900 min *while held* at 80 K;”

2. In Fig S2 (supporting information), the optical image of the crystal at 260 K looks really crack free with clear crystal surface. However, no single crystal data is available at RT. As authors have not performed any thermal analysis, it is remained unclear how the crystal behaves at room temperature.

An optical image was acquired at room temperature (295 K) but was not included in the original Supplementary Information because 260 K is the upper limit in temperature when the oil that encloses the crystal is sufficiently viscous to firmly fix the crystal to the backing substrate. The contrast of the image at 295 K is therefore slightly different, although this issue appears to be minimal upon checking, following this request. Accordingly, we have now added the optical image of the crystal taken at 295 K to Figure S3 (originally Figure S2) by substituting out the optical image of the crystal at 130 K since that seems to be the least important in terms of showing changes). This optical image at 295 K shows that the crystal is crack free at room temperature, in common with the 260 K image upon which this reviewer has already remarked. A note to the figure caption has been added to explain this to the reader.

We note that this reviewer comments specifically about Figure S3 (originally Figure S2) (*i.e.*, the optical images in the Supplementary Information) rather than the more selective optical imagery shown in Figure 3 of the main paper. We thus assume that this reviewer is content that we do not alter Figure 3 since the reader can of course look at the fuller temperature series in the Supplementary Information. Nonetheless, we have added a comment in the main paper within the caption of Figure 3 about the practical complication of the 295 K data vis a vis the oil fixing issue and its slight impact on the image contrast. We hope that this clarifies the situation.

3. The isomerisation occurs with a clear bond breaking/making around the SO₂ with respect to metal ion. Do the space around the SO₂ group is changing and impacting any lattice rearrangement within the crystal? The surface nanoscopic crystal strain can be easily correlated to the inner structural changes.

This comment refers to what is described in this field as the ‘SO₂ reaction cavity’ that defines the shape and volume of space that the SO₂ ligand has room in which it can manoeuvre within its surrounding crystal lattice. For further details, see our already cited paper about the SO₂ reaction cavity in a previous study of a compound in this series of materials:

Phillips, A. E., Cole, J. M., d’Almeida, T. & Low, K.S. Effects of the reaction cavity on metastable optical excitation in ruthenium-sulfur dioxide complexes. *Phys. Rev. B* **82**, 155118 (2010).

The single-crystal x-ray diffraction results of the two light-induced crystal structures of **1** at 90 K and 100 K show that the lattice arrangement within the crystal is maintained upon light exposure. Yet, the very large (0.4-1.0 Å) changes in unit-cell parameters, *a* and *b*, of the crystal structure upon light exposure (see Table 1 of the paper) evidence that severe crystal lattice strain is caused by the formation of either photoisomer. We did note this in the original

manuscript but given this query, we wish to clarify the matter – accordingly, we have added the following sentence (highlighted in bold italics) to the relevant text of the paper that explicitly mentions the SO₂ reaction cavity to help contextualise the phenomenon. Thank you for pointing out this matter.

“This high degree of unit-cell distortion indicates that the size and shape of the SO₂ reaction cavity, in which the SO₂ ligand has room for manoeuvre within its surrounding crystal lattice, is put under severe crystal-lattice strain owing to the formation of either photoisomer. In turn, this large crystallographic change in **1** presumably accounts for the severe level of microscopic strain that was observed in the crystal as a result of light irradiation, via optical microscopy (Figure 2(a)).”

4. Is there any change in intermolecular interaction during the photo isomerization (η^1 -SO₂ to η^1 -OSO to η^2 -(OS)O isomers. How does these intermolecular interactions contribute to the stability to system? A detailed theoretical study may be added.

Yes. On the one hand, light exposure depletes two strong steric forces between ammine ligands and counterions. On the other hand, the reaction cavity is stretched upon light exposure, which places crystal-lattice strain upon **1**. These are the salient interionic interactions.

We realised these findings by constructing Hirshfeld surfaces of the η^1 -SO₂ dark-state, η^2 -(OS)O and η^1 -OSO light-induced crystal structures of **1** which have been added to the Supplementary Information of the paper in a new section, C1. Therein, these three Hirshfeld surfaces are all viewed looking down the crystallographic axis, *a*. The red, white and blue regions show positive, neutral and negative isoenergies. The images were generated via CrystalExplorer17 (2017). <https://hirshfeldsurface.net>. The light-induced depletion of two strong steric forces between ammine ligands and counterions that are indicated in red by the two black arrows annotated to the dark-state Hirshfeld surface. The blue volume of the Hirshfeld surface for the dark-state species (691 Å³) contracts to 680 Å³ with light exposure at 90 K while its surface area (554 Å²) increases to 559 Å²; *i.e.*, the Hirshfeld surface is stretched upon light exposure which strains the reaction cavity.

5. Only two meta stable isomers (light 90K, light 100k) are successfully isolated by SXCRD analysis. However, SCXRD may not able to isolate low stable meta-states. Additional experiments or a detailed theoretical study may add more insight of it. A variable temperature PXRD may be useful also.

Experience suggests that we are unlikely to see any metastable states than we have already seen, *cf.* references 27-45 cited in our paper. Nonetheless, the reviewer is right to not limit their view to previous work, as another metastable state could in principle exist even if unlikely.

We have found that our best recourse to exploring temperature and light space to discover light-induced metastable states of these complexes is our use of single-crystal optical absorption spectroscopy. Indeed, we have used this spectroscopic method in the current study to explore the subject complex over a range of temperature conditions, with 505 nm light stimulation. Previous work on varying the wavelength of visible light in this series of complexes [40,41] has shown that we are best to photo-stimulate the sample near (but not at) the peak of the optical absorption spectrum for the dark-state SO₂ species and green light has proven to be optimal; while the application of red light will have a similar effect to warming the sample, in that the photoinduced state reverts to the dark-state SO₂ species.

Accordingly, our herein presented single-crystal optical absorption spectroscopy measurements conducted as a function of temperature offer a means by which other metastable states can be identified if they are to exist, and none were found in this study (as expected, but nonetheless checked). **See Figure 2 of the paper and section A of the Supplementary Information for details; note that we have added more data to the Supplementary Information that show results on this complex on a wider range of temperatures, i.e. sampling a wider chemical space.** We hope that this clarifies the query from this reviewer.

Regarding the two alternative methods that this reviewer suggests, it is perhaps helpful to add that while these may indeed be suitable methods for studying other classes of materials, they were not suited to studies on this particular family of compounds for the following reasons:

(1) A detailed theoretical study was not considered to be credible given that this is an exclusively single-crystal structural phenomenon. As such, electronic-structure calculations need to be performed using plane-wave density-functional-theory (DFT) calculations which have been shown to be extremely challenging. *cf.*

Jain A, Cole JM, Vázquez-Mayagoitia Á, Sternberg MG. Modeling dark- and light-induced crystal structures and single-crystal optical absorption spectra of ruthenium-based complexes that undergo SO₂-linkage photoisomerization. *J Chem Phys.* (2021) 155(23), 234111.

In this work, we managed to use such calculations to replicate crystal structures of compounds in this series of materials but our results were not sufficiently accurate to model the corresponding optical spectra which realistically precludes their credibility if using them to predict new metastable state crystal structures.

Alternatively, one can perform gas-phase calculations on a supercell that comprises a sufficiently large cluster of ions that it includes the full SO₂ reaction cavity of the compound; yet, this approach also lacks credibility given that gas-phase calculations do not represent the solid-state phenomenon at hand which is key to this light-matter phenomenon. Besides, our DFT work cited above, which also attempted this alternative form of DFT calculations, found that our plane-wave results were more reliable than these molecular-orbital calculations, as one might expect. Furthermore, it is helpful to remember that density functional theory (DFT) models assume an electronic structure at 0 Kelvin.

(2) Powder X-ray diffraction (PXRD) is also no go when studying these single-crystal optical actuators because microcrystalline samples of this series of complexes are opaque to visible light, *i.e.*, at the wavelength of interest. Indeed, we have previously crushed single crystals of our samples to form powders when attempting such experiments, but the effect is like smashing a glass window, in that visible transparency through the material is completely lost.

6. The observed surface nanoscopic crystal strain is purely qualitative and has not quantified. A detailed nanoindentation on the crystal surface may help in this regard and can be added.

Thank you for this interesting suggestion. We understand that a detailed nanoindentation on the crystal surface may help, although this would represent an entire study within its own right. It may help to clarify that the herein presented low-temperature *in-situ* light-induced atomic force microscopy proved to be extremely challenging to perform; it took several attempts by a team of highly technically experienced scientists and months of work to succeed in realising this aspect of the work presented; indeed, this is presumably why our AFM results herein represent such a

rare example of applying AFM to such a field of study. We agree that this is an interesting idea, although we would like to advocate that this is out of scope for this paper, and suggest that we will be happy to consider this advice for the future. Thank you for raising this idea.

Minor comments

1. Improve the introduction section with a broad (types) isomerization process exists in inorganic systems where meta stable exists and have successfully isolated.

Thanks for this advice. We were conscious on the limit of references stipulated by the guidance notes of *Nature* publications, but given this response, we have now broadened our reference base to mention six additional studies in the introduction section that span a wider range of chemistry on single-crystal photo-actuators of inorganic systems where metastable states exist and have been successfully isolated.

2. The resolution of ORTEP in figure 1 looks very poor. Change to a better one.

Thanks for this suggestion, although we have checked the version downloaded from the *Nature* portal and it looks fine to us. We have therefore maintained the original artwork for Figure 1. Perhaps something went wrong in the pdf translation for this reviewer? (No other reviewer has mentioned such an issue).

Reviewer #3 (Remarks to the Author):

The article 'Ternary molecular switching in a single-crystal optical actuator with correlated crystal strain' is dedicated to one of the popular modern topics of mechanical response to chemical transformations in solids (about 40 years should have passed since the first quite deep studies of photomechanical and chemomechanical phenomena, so that the community began to show massive interest in these issues, at least at the phenomenological level). This paper is precisely an example of research useful for the development of this direction. I consider the attention to the temperature dependence of the Ru-SO₂ complex photoisomerisation pathways, as well as the attention to the factors of changes in structural parameters caused by the transformation, to be a particularly valuable feature (and result) of the study. Therefore, I believe that the paper can be published in *Nature Communication*.

Thank you for this positive recommendation of our work for *Nature Communications*.

Nevertheless, some changes are needed - there are both formal comments on the text and conceptual questions, the answers to which would improve the article.

1. There seems to be a number of inconsistent designations for the bidentate type of Ru-(OS)O coordination in the text. In different parts of the text there are designations of the form: h1-(OS)O, h2-O(OS) and h2-(OS)O.

Thank you for spotting these inconsistencies. We have now performed a comprehensive search of -(OS)O, -O(OS) and -(OS)O notation used throughout this manuscript and have found two typos (both in the introduction section) that we have now corrected.

2. There is some disagreement about the light wavelength used in the AFM experiment. The description of the technique says 510 nm, but the caption to Figure 4 says 505 nm.

Thank you for spotting this inconsistency. The AFM experiment did indeed employ 510 nm light; the caption to Figure 4 contains a typo which we have now corrected.

3. The description of the Figure 4 in the last paragraph on page 8 (starting with ' Figure 4a shows the crystal surface...' and ending with '...just over 4 μm in width (Figure 4d).') sounds almost completely incomprehensible. Perhaps these features in the figures mentioned there need to be labelled somehow?

Thank you for this feedback. We had originally kept very brief the number of images in both Figures 4 and 5 of the main text of this paper and relied on the Supplementary Movies S4 and S5 to clarify these features. However, this feedback shows that we should actually present a much larger series of images within the manuscript and use annotations to guide the reader to each feature therein, as well as support these figures with Movies S4 and S5 that encompass the whole sequence of structural changes.

Accordingly, we have revised both Figure 4 and 5.

Regarding Figure 4:

Our revised Figure 4 (now 12 images instead of 4) should make the original text much clearer, although we have also modified the text in line with the changes in Figure 4 to elucidate the structural descriptions.

We have also responded to each specific point raised by this reviewer in a point-by-point fashion below (a-d), to deal with our clarified description of each feature in Figure 4.

Regarding Figure 5:

Given this feedback on Figure 4, we realised that it would be best to also add more images to Figure 5 in this revised manuscript, as this will better illustrate the progressive closing up of the main channel (microscopic crack) in the crystal as it is warmed up above its metastable temperature and onwards to 295 K. Accordingly, Figure 5 now contains 16 images instead of 4. Minor updates to the associated text for Figure 5 have been made in line with these changes.

Regarding associated Supplementary Movies S4 and S5:

We would also like to mention that we have updated the fonts of the cross-sectional axes and numerical labels for both Supplementary Movies S4 and S5 during the course of these revisions, so that they are clearer to the reader.

a. Where is the zig-zag pattern in Fig. 4a?

Figure 5a has now been annotated with two parallel sets of yellow dotted lines that enclose the central line-of-sight through the zig-zag structural pattern to guide the reader to a zig-zag motif. A textual description of this image annotation has been added to the caption of Figure 5.

b. Where is the vertical line at $x = -1.6 \mu\text{m}$ in Fig 4b?

Figure 5e has been annotated with an arrow at the top of its image to help the reader identify the location of this feature. A textual description of this image annotation has been added to the caption of Figure 5. We have also renamed this 'vertical line' as a 'vertical stripe' in the text as

that is a better description of the feature, and we have now used the central point of this stripe (-1.3 μm) to define its location.

c. Where are the two vertical lines in 4c?

Figure 5l has been annotated to help the reader identify these features using four arrows: two arrows on top to identify the location of the two vertical stripes (which represent channels in the crystal surface that lie 2 μm apart) and two arrows at the bottom to indicate their width of 1.3 μm . A textual description of these image annotations has been added to the caption of Figure 5.

d. What was the point of choosing the section line at $y=0$? It seems to run along an area of low information, while the lower half of the image shows the appearance of deep crack-like areas.

The $y = 0$ line was chosen as it allows one to observe structural features across all AFM images over a good dynamic range, while avoiding any human bias e.g. a temptation to centre deliberately on the most intense features; a cross-section half-way down the image seems to be neutral.

We have added the above sentence to the revised manuscript to clarify this matter.

4. The use of the terms microscopic and nanoscopic strains seems inappropriate. What is meant in the text are defects (cracks or pores) of microscopic and nanoscopic scales, so they should be called that. The formations seen here on crystal surfaces in the optical microscope and AFM are defects (such as cracks or pores) due to deformation caused by the isomerisation. Instead, the authors refer to the defects as strains, which is incorrect.

Thank you for pointing out this subtle but important point. We were focused on the light-induced crystal-lattice strain being the cause of the formation of microscopic and nanoscopic cracks in the crystal. We have now worked through every instance that 'strain' is mentioned in the manuscript and either clarified its meaning by highlighting that that strain is the underpinning cause of the observed cracking phenomenon, or changing the word 'strain' for 'cracking/cracks'.

5. Fracture is visible on the optical microscopy images, even on the surface of the original crystals. This may mean that the crystals were fractured under irradiation by scattered daylight, even before they were irradiated in the experiment. This is not surprising since the absorption spectrum occupies the entire visible part of the spectrum and the transformation causes a fairly significant compression by $\sim 10\%$ along the b crystal axis. If this is true, one could say that all the photomechanical phenomena observed here are related to the opening (in the isomerised states) or collapse (in the stable isomer state) of the crack cavities. This raises two questions:

a. Did the authors not try to minimise the illumination of the crystals during their growth and preliminary operations on them?

Yes, we did. The crystals precipitate directly from the reaction mixture upon cooling it down. We enshrouded the reaction vessel with aluminium foil throughout the experiment and minimised light entering the fume hood used for the synthetic procedure. The crystals were then washed and subsequently stored in complete darkness.

Regarding the crystal handling during the materials-characterisation experiments, the windows of the diffractometer were covered with aluminium foil **when performing the low-temperature X-ray diffraction experiments**. The crystals were exposed to light when using an optical microscope to mount the crystal onto a loop and then onto the diffractometer. However, the

crystals only photo-isomerise at low temperature (of the order of 90 or 100 K), so this procedure should not have affected the crystal quality because the crystal is exposed to cryogenic conditions only once it has been mounted onto the diffractometer as it is equipped with an open-flow cryostream. Most of this procedure was summarised in the originally submitted paper, in section: *Materials and Methods – Materials Characterisation – Dark and Photo-induced In-situ Single-Crystal X-ray diffraction of 1*.

Given this feedback, we have added to this section the point that:

“The windows of this diffractometer were covered with aluminium foil ensure a controlled light-induced experiment”

When we perform the single-crystal optical absorption spectroscopy experiments, multiple crystals are simultaneously placed in the cryostat before cooling so that the cryostat is only open once before we vacuum down. We focus on photo-inducing one crystal at a given time via optical focusing, but the other crystals in the periphery field-of-view may absorb a little bit.

When aligning a given focused crystal in preparation for measurements, we place the microscope on its lowest power during this period. It is also standard for us to add high optical density filters between the lamp and the condenser if we observe any change in the crystal during this period, although low powering the microscope was sufficient for this experiment.

We also employ a visible bandpass filter (Schott: BG40) to remove red light from the experiment, as that is known to deactivate photo-isomers in this series of complexes.[40] See technical specifications spectrum of this filter on:

<https://www.schott.com/shop/advanced-optics/en/Matt-Filter-Plates/BG40/c/glass-BG40>

These provisions are standard in our custom-built single-crystal optical absorption spectroscopy set up [41], as have been borne out by a decade of experience in learning to work with this series of complexes on this apparatus in a light-controlled fashion.

We have already noted the key elements of this information in the section of the paper: *Materials and Methods – Single-crystal optical-absorption spectroscopy and microscopy of 1*.

Nonetheless, we hope that this additional information for this reviewer provides helpful clarification.

Regarding the low-temperature *in situ* light-induced atomic force microscopy experiment, the crystal had to be mounted in light conditions, but we kept the crystal at room temperature during this period. The room lights were extinguished before cooling the crystal cryogenically. A note to this effect has now been added to the AFM part of the Materials and Methods – Materials Characterisation section of the paper, according to:

“The room lights housing the AFM instrument were then extinguished before the sample was cooled to...”

b. Did the authors not observe fracture-free crystals of smaller size, hypothetically less than 5-10 μm thick? If so, such crystals could demonstrate reversible bending or even twisting into a spiral (since the structure has triclinic symmetry) instead of fracture.

A range of crystals with different sizes were tested but all showed the same phenomena. Some of the crystals in the Supplementary Information Movies were also shown after several cycles of

light-induction and thermal warming. For example, Movie S3 shows same phenomena via the warming up cycle of a light-induced crystal whose thinnest dimension is *ca.* ½ the thickness of the crystal which we have highlighted in the main manuscript. This thickness is the key criterion here because the crystal is exposed to light through the thinnest dimension of these crystals in order to maximise the optical penetration depth of the incident light.

If the reviewer looks very carefully in Figure 3 (and Figure S3), they can also see a very small crystal (of the order of size that this reviewer mentions) lying on the right-hand side of the main crystal: see a needle-shaped crystal located just below the dark crack in the oil film. This crystal also shows light-induced cracks that run along its longest dimension and which disappear once it has been warmed to room temperature.

Thanks for this interesting query. We hope that this helps to clarify the matter for this reviewer.

6. What are the directions of the crystal axes *a*, *b* and *c* in crystals? This information should be available after single-crystal XRD. It could help in interpreting the observed effects. From the orientation of the cracks on the surface visible in Figure 3, one can hypothesize that *b* axis is almost perpendicular to the long axis of the crystals – structure compression by ~10% along *b* axis should cause fracture in the surface layers, leading to the formation and opening of cracks perpendicular to *b*.

Yes, that is correct.

Unit-cell parameter, *a*, runs along the longest crystal axis - where the crystal expands upon application of light; this stands to reason because the crystal can be seen to crack at its base in a few places as if its entire base is very slightly buckling up (with striations seen up on the crystal surface that traverse the main length of the crystal); these cracks disappear upon warming. In a general respect, *a* is the shortest unit-cell parameter which often corresponds to the longest dimension in a crystal habit.

Unit-cell parameter, *b*, runs along the width of the main face of the crystal (*i.e.* perpendicular to the long axis of the crystals as suggested by this reviewer). *b* is contracting by a large amount which means that cracks will appear, as the entire crystal is not shrinking in other directions.

Unit-cell parameter, *c*, runs along the thinnest length of the crystal (which makes sense as it shows the least change upon application of light, and it is the longest unit-cell dimension which often corresponds to the shortest length in a crystal habit).

These crystallographic directions are akin to those seen in the optomechanical ‘peeling’ effect that we observe in a crystal of another complex in this series of materials (*cf.* [38] in this paper), albeit the sense and order of contractions and expansions differed in that case, thereby affording a different microscopic manifestation.

We agree that this information would help to interpret the observed effects. Thus, we have added the following sentence to the paper at the end of the results section about the crystal structures where we discuss the large changes in unit-cell parameters upon light exposure:

“Indeed, changes in the crystallographic unit-cell parameters *a*, *b* and *c* of **1** run along the longest, middling and shortest sides of its crystals, respectively. The very large crystallographic contraction along *b*, with a lack of compensating contractions along other unit-cell axes, will essentially cause microscopic cracks to form in the crystal perpendicular to the *b*-axis, *i.e.*, cracks that run along the length of the main face of the crystal (*cf.* Fig. 2(a)).”

7. The cracks on the crystal surface are due to the inhomogeneity of photoisomerisation, whose scale is determined by the light absorption depth in this substance. Since the visible fracture scale is ca 10 μm , one can assume that the absorption depth is also close to this scale. The depth of absorption is a very important property that determines the type of the mechanical response, which can be understood from [7] on the reference list, as well as from other works by these authors. If the depth is on the order of the crystal thickness, the crystal won't crack, but will bend or twist into a spiral, depending on the symmetry. If the crystal is significantly thicker (e.g. by an order of magnitude or more), the transformation will typically lead to fracture, since the strain inhomogeneity is too great for the crystal to simply bend without fracture. Moreover, if the thickness of the crystal is much greater than the absorption depth, it may take a long time to approach 100% transformation of the entire crystal volume (irradiation intensity and transformation rate decrease exponentially in deep regions of the crystal). This raises a number of other questions:

a. Did the authors estimate the absorption depth (from direct measurements at their facility or from absorbance obtained from UV-Vis spectra of the substance in solutions)? How is it related to the absorption depth?

The single-crystal optical absorption spectroscopy experiments are carried out in transmission mode, so by definition the light does pass through the crystal (else we wouldn't see any results).

As can be seen from the 90 K dark-state optical absorption spectrum of the crystal of **1** in Figure 2, its optical absorbance (OA) is ca. 0.35 at the light-induced wavelength of 505 nm. Thus, the crystal will be absorbing just over 50% of light, given the OA profile of the optical absorption spectrum of **1** (labelled as 90 K, 2h 505 nm light in Figure 2).

Note that this represents non-linear optical absorption behaviour that does not comply with the Beer-Lambert law. This is what we would expect given that our measurements are acquired on the single-crystal, *cf.* our published explanation of this matter in section 3.1.1.4 of our paper [41] and references therein, *i.e.*,

Duyens, L. N. M. The flattening of the absorption spectrum of suspensions, as compared to that of solutions. *Biochim. Biophys. Acta* **1956**, *19*, 1–12, DOI: 10.1016/0006-3002(56)90380-8

Corbett, D.; Warner, M. Linear and Nonlinear Photoinduced Deformations of Cantilevers. *Phys. Rev. Lett.* **2007**, *99*, 174302, DOI: 10.1103/physrevlett.99.174302

Corbett, D.; Warner, M. Bleaching and stimulated recovery of dyes and of photocantilevers. *Phys. Rev. E: Stat., Nonlinear, Soft Matter Phys.* **2008**, *77*, 051710, DOI: 10.1103/physreve.77.051710

Thereby, the key information that pertains to this query can be derived directly from Figure 2 in this paper; and at the end of our paper section: *Materials and Methods – Materials Characterisation – Single-crystal optical absorption spectroscopy and microscopy of 1*, we have already referred the interested reader to further details via our technically-focused paper that explains this non-linear behaviour in this series of complexes ([41] in this manuscript).

Thus, we have not added detail to the paper on this point, but we hope that this response helps to clarify the situation for this reviewer.

b. What exactly is meant by "100% conversion" in the text: near 100% conversion of the entire volume of the crystals (~100 μm thick), or 100% conversion in the surface layer of the crystals? If, as suggested above, the absorption depth is ~10 μm , then the typical photon flux of the 3 W LED used in the experiments here is almost sufficient to achieve almost complete conversion to a depth of 100 μm in 2 hours (assuming a high quantum yield of the isomerization). However, if the absorption is much higher or the quantum yield is low, complete transformation of the entire volume may not be achieved.

We are confident that the crystals of **1** achieve 100% photoconversion at the temperatures and light-induced conditions stated in this paper, as they have been verified by light-induced crystal structure determinations. *i.e.* the 'gold standard' of advanced materials characterisation (a bulk characterisation technique and thus the results are the average over the entire crystal *i.e.* categorical evidence that the whole crystal has been 100% photoconverted).

Thereby, the atomic positions of every atom in the SO_2 ligand in each photoisomer have been independently refined unambiguously with occupancy factors of **1**, *i.e.*, 100%-ordered SO_2 species in three crystal-structural configurations are presented: the $\eta^1\text{-SO}_2$ isomer in the dark-state crystal structure at 100 K (Figure 1(left)), the $\eta^1\text{-OSO}$ photoisomer in the light-induced crystal structure at 90 K (Figure 1(middle)), and the $\eta^2\text{-(OSO)}$ photoisomer in the light-induced crystal structure at 100 K (Figure 1(right)).

We also refer to the above note about non-linear absorption effects for further clarification.

Many thanks for the interest shown in the review of our paper.

POINT-BY-POINT RESPONSE TO THE REVIEWER COMMENTS

Reviewer #1 (Remarks to the Author):

The authors appropriately replied the questions raised by the three reviewers and made proper revisions in their revised MS. The results are solid and this work will attract a wide readership of researchers involved in photochemistry, photo-driven devices and soft robotics, etc.. I believe this revised manuscript could be accepted and published in this journal without additional corrections.

Authors' response: Thank you for supporting the view that this revised manuscript is now suitable for publication in Nature Communications.

Reviewer #2 (Remarks to the Author):

The authors answered most of the questions raised by the three reviewers. The revised version of the manuscript has been significantly improved after incorporating the reviewers feedback. I could recommend its acceptance in the present form.

Authors' response: Thank you for supporting the view that this revised manuscript is now suitable for publication in Nature Communications.

Reviewer #3 (Remarks to the Author):

While I am in disagreement with a number of the authors' conclusions and assertions (which I will discuss in further detail below), I would like to express my gratitude to them for their meticulous consideration of all the issues raised. Furthermore, I would like to note that the amendments made to the text make it clear and suitable for publication in Nature Communication.

As a point for discussion, which does not affect my decision on the possibility of publishing this manuscript, I would like to highlight facts that may have led to a potential misunderstanding of the nature of some of the results.

Authors' response: Thank you for supporting the view that this revised manuscript is now suitable for publication in Nature Communications. We respond below to the final two queries by this reviewer that concern arguments that we made in our original point-by-point response; accordingly, these final clarification points do not change the manuscript or SI. We are grateful to this reviewer for the further discussion on these points.

Reviewer 3 comment 1:

Firstly, the authors' assertion that the absorption is not linear and does not obey the Beer-Lambert law (as well as similar assertions in the reference Corbett, D.; Warner, M. Linear and Nonlinear Photoinduced Deformations of Cantilevers) is erroneous. A potential misunderstanding of the Beer-Lambert law itself is indicated in Phys. Rev. Lett. 2007, 99, 174302, DOI: 10.1103/physrevlett.99.174302. In the cases under consideration, it would appear that the authors are suggesting that due to the alteration in the chemical composition of the phase (the occupation of positions by different isomers) caused by isomerisation, the extinction of the substance undergoes a change and, in general, becomes spatially non-uniform. This, in turn, results in the law of change in the intensity of healing with depth becoming non-exponential. Nevertheless, this does not imply that the principle of linear absorption, which forms the basis of the Lambert-Beer law, is no longer fulfilled. Its differential form remains linear in intensity: $dI/dx = -a(C)I$. The deviation from the exponential form $I = I_0 \exp(-a_0 \cdot x)$ arises due to the non-uniformity of the absorption coefficient $a(C)$ (if the concentration of isomers $C(x)$ is non-uniform, then $a(C(x)) \neq \text{const}$), rather than due to the non-linearity of the absorption process itself. Nonlinear absorption is a completely separate phenomenon, characteristic of nonlinearly absorbing substances (processes such as two-photon absorption, for example, leading to quadratic contributions to the absorption in intensity), and arising, in any case, at very large absorption coefficients and high intensities.

Authors' response: This query refers to our original authors' response to reviewer 3, point 7.a. Accordingly, we suggest the following revision to this point:

a. Did the authors estimate the absorption depth (from direct measurements at their facility or from absorbance obtained from UV-Vis spectra of the substance in solutions)? How is it related to the absorption depth?

The single-crystal optical absorption spectroscopy experiments are carried out in transmission mode, so by definition the light does pass through the crystal (else we wouldn't see any results).

As can be seen from the 90 K dark-state optical absorption spectrum of the crystal of 1 in Figure 2, its optical absorbance (OA) is ca. 0.35 at the light-induced wavelength of 505 nm. Thus, the crystal will be absorbing just over 50% of light (assuming a negligible change in reflectivity), given the OA profile of the optical absorption spectrum of 1 (labelled as 90 K , 2h 505 nm light in Figure 2).

We agree with reviewer 3 that the photoisomerisation process results in a

“deviation from the exponential form $I = I_0 * \exp(-\alpha_0 * x)$ [that] arises due to the non-uniformity of the absorption coefficient”.

We appreciate that we caused confusion by use of “non-linear” terminology to describe this deviation in our original response. We had not intended to refer to different optical processes such as two-photon absorption. Indeed, we agree that such processes are entirely different.

We hope that this further explanation clarifies this point. Many thanks for raising it.

Reviewer 3 comment 2:

Secondly, optical absorption values in Fig. 2, which do not exceed approximately 0.8 for 505 nm radiation, provide clear evidence that the crystal thickness does not exceed $1.6 = -\ln(0.2)$ times the characteristic absorption depth. It is therefore unlikely that the deformations observed under such conditions will result in significant stresses within the crystal, nor will they be capable of causing fracture. Instead, the crystal will experience macroscopic deformation that is almost entirely consistent with the inhomogeneous distribution of isomers along the thickness, arising at an intermediate stage of transformation and residual stresses. For further verification of this claim, the reader is directed to the mentioned work of Corbett and Warner (2007) and mostly to ref. [10] of this work (M. Warner and L. Mahadevan, Phys. Rev. Lett. 92, 13430). However, the absorption of red light at ~650 nm is markedly high, which could potentially lead to significant inhomogeneity in the transformation at a later stage, capable of inducing high tensile stresses in the b direction and causing fracture. Related to this was my question about preventing the crystals from being illuminated before the experiment. My assumption, consistent with the results observed in the work, remains that authors always worked with defective crystals and simply observed crack opening or closing under the action of the resulting deformations.

Authors' response: The following reviewer 3 query refers to our original authors' response to reviewer 3, points 5.a. and 5.b. Accordingly, we suggest the following revision to this point:

We take on board these points. It may be helpful to clarify a few practical matters:

(1) We don't see this fracture effect in any of the papers on this family of complexes that we have already published. Indeed, our only published work to date that shows single-crystal optical actuation in this family of complexes is an example of a 'peeling' crystal [reference 38 of the manuscript].

(2) The crystals in our experiments are necessarily pinned to the substrate owing to the practical matter of keeping them stationary throughout the experiment. As we detailed in the

Methods section of the paper, we employ frozen oil to fix the crystals in our optical microscopy experiments and epoxy to fix them for the AFM experiments. We therefore cannot consider stresses within the crystal in isolation from the external forces caused by pinning down the crystals for the experiments. Therefore, it is not as simple as saying that the crystal should bend instead of fracture owing to its characteristic absorption depth.

(3) As per our response to reviewer 3's point 5.a., we took great care in preventing the crystals from being illuminated prior to the experiment. It may be worth adding one point about the optical absorption spectroscopy experiment which we didn't state previously – which is that we entirely block the microscope light to the sample during the 505 nm illumination of the crystal.

Taken together, we hope that these extra points clarify the situation.